# CAUSAL PATH TRACING IN TRANSFORMERS

## ABSTRACT

We propose a causal path tracing framework to understand how information causally flows through the internal structures of transformers for a given decision. By unfolding each block into a causal graph of path nodes and applying a *minimality-based subset search*, our method identifies all possible causal paths within each block, with polynomial-time complexity on average. Furthermore, we demonstrate the reliability of a *union-based causal path reference strategy*, enabling efficient and reliable causal tracing throughout the model. The key contributions of this work are: (1) an automated, efficient framework for causal path tracing that exhaustively searches paths along direct dependencies; (2) theoretical and empirical validation demonstrating exhaustive search with polynomial-time complexity on average; (3) experimental findings showing that self-repair effects occur far less frequently along the identified causal paths, that certain paths are uniquely activated for specific classes, and that the traced paths are both accurate and faithful.

## 1 INTRODUCTION

With the success of transformers (Vaswani et al., 2017) across language and vision, interest has grown in understanding their internal mechanisms beyond their black-box nature, especially to enable safer deployment in high-stakes applications such as healthcare, law, and education. Mechanistic interpretability aims to identify specific components within the model, such as attention heads or MLPs, that contribute to its behavior. Building with mathematically grounded circuit discovery in simplified settings (Elhage et al., 2021), recent efforts have incorporated Pearl's causal theory (Pearl, 2009), employing ablation-based interventions to trace which parts of the network support outputs.

Depending on the granularity of analysis, prior work can be classified into: *node-level patching* (Vig et al., 2020; Wang et al., 2022; Meng et al., 2022a;b; Heimersheim & Janiak, 2023; Zhang & Nanda, 2024), which identifies the role of individual input features; *edge-level patching* (Conmy et al., 2023; Bhaskar et al., 2024; Syed et al., 2024; Hanna et al., 2024), which examines the influence of neighboring feature pairs with direct computational dependencies; and *path-level patching* (Chan et al., 2022; Hanna et al., 2023; Nanda et al., 2023), which investigates the contribution of distant feature pairs connected through multiple accumulated dependencies.

Recent work (Rushing & Nanda, 2024) has shown that ablation-based methods often fail to estimate true causal effects due to self-repair (or backup behavior, see Appendix Section G.2), where later components compensate for earlier ablations. This implies that, when unablated components lie between the target and the decision, internal explanations may be misattributed. To address this, one solution is to iteratively evaluate each component conditioned on priorly identified causal components along direct computational dependencies; here, we refer to as *causal referencing*. However, prior node- or edge-level approaches cannot fully support causal referencing over all relevant combinations; though sequentially feasible, it remains inaccurate. In contrast, path-level approaches can in principle support this, but due to their combinatorial complexity, existing studies typically rely on hypothetical tests (Chan et al., 2022) or assess a single subpath manually (Hanna et al., 2023; Nanda et al., 2023), making it infeasible to capture a full explanation for a decision in time. (Table 1)

To address this obstacle, we propose an automated and efficient framework for tracing causal paths given a decision. Specifically, we begin by unfolding all possible paths within each block of a transformer, interpreted as a causal graph, into path nodes. Then, by introducing a minimality-based subset search strategy for identifying all possible causal path node combinations per block, we reduce the inherently exponential complexity to polynomial time on average. Furthermore, to enable efficient block-wise tracing, we demonstrate that referencing the union of causal paths identified in preceding

| Approach | Patching | Path Tracing for Decision | |
|---|---|---|---|
| | | **Feasibility** | **Reliability** |
| Vig et al. (2020); Wang et al. (2022); Meng et al. (2022a;b) Heimersheim & Janiak (2023); Zhang & Nanda (2024) | Node | ✓ backward chaining only | ✗ no causal referencing |
| Bhaskar et al. (2024); Syed et al. (2024); Hanna et al. (2024) | Edge | ✓ backward chaining only | ✗ no causal referencing |
| Chan et al. (2022) | Path | ✗ hypothetical only | ✓ full coverage |
| Hanna et al. (2023); Nanda et al. (2023) | Path | ✗ manual subpath | ✓ full coverage |
| **Ours** | Path | ✓ polynomial on average | ✓ full coverage |

Table 1: **Comparison of patching methods for path tracing in a given decision.** Feasibility refers to empirical applicability for a given decision; reliability to its theoretical guarantees. Backward chaining refers to simply concatenating the tracing results obtained independently at each block without causal referencing (Definition 4).

blocks not only makes this feasible but also ensures reliability. We also note that, as in prior work, the decision we trace corresponds to the model's predicted class in a classification setting.

Our approach reveals that self-repair occurs primarily outside the identified causal path; thus, the path contains information essential to the decision and not easily replaced, reflecting its critical role. Moreover, we found that there exist causal paths uniquely associated with specific classes. These paths are activated only for their corresponding classes, serving class-specific roles within the model. Taken together, our results show that the proposed method faithfully and accurately explains model behavior under empirical evaluation.

## 2 METHODOLOGY

In this section, we formalize our causal framework and introduce the causal tracing method for a given decision. Section 2.1 examines whether a transformer can be formally defined as a causal graph, and formalizes what constitutes a causal path in our framework. Section 2.2, to assess whether an internal structure is causal by isolating it through intervention, introduces what makes an intervention sufficient in our framework. Section 2.3 specifies how we unfold each transformer block into nodes in a causal graph. Section 2.4 then introduces our method for identifying causal paths by performing interventions efficiently within each block. Finally, Section 2.5 extends the procedure of Section 2.4 to the full model, ensuring that all possible path combinations for a given decision can be evaluated.

### 2.1 CAUSAL FRAMEWORK AND DEFINITIONS

To proceed, we introduce the definitions used throughout this work. Our goal is to reveal and explain internal components for decision by efficiently tracing causal paths. To enable this, following Pearl's causal theory (Pearl, 2009), we interpret the transformer as a causal graph, as formalized in Definition 2. Based on this interpretation, we define the causal path for a given model decision through Definitions 1 and 3 to 7, where Definition 5 is adapted from the Halpern–Pearl definition of actual causality (Halpern & Hitchcock, 2011; Halpern, 2015).

**Definition 1** (Computational Dependency). *A computational dependency is a transformation from one node to another in the model's computational graph. If a transformation cannot be decomposed into any intermediate transformations, we call it a direct computational dependency. In our formulation, each direct computational dependency corresponds to a **directed edge** between the two nodes.*

**Definition 2** (Transformer as Causal Graph). *We say that a transformer is a **causal graph** $\mathcal{G} = (\mathcal{V}, \mathcal{E})$, where each node $v \in \mathcal{V}$ denotes an internal component (namely, unfolded path terms as described in Section 2.3) and each edge $v_i \rightarrow v_j \in \mathcal{E}$ indicates a direct computational dependency, if it satisfies the following conditions:*

*(2.a) **Directed Acyclic Graph:** Its internal computation proceeds layer by layer in a forward direction without cycles, which naturally forms a directed acyclic graph structure.*

*(2.b) **Markov:** Each node depends only on its parent nodes. Once its parents are fixed, it is conditionally independent of its non-descendants.*

*(2.c) **Causal Sufficiency:** All nodes involved in its internal computation are observable, with no latent confounders or hidden common causes among nodes.*

*The direction of all edge follows the model's forward computation: edges point from the input side node (parent) to the output side node (child). In addition, the **parent node set** of a child node is defined as the possible set consisting of its parent nodes. Any node or node set that can be reached by following one or more edges from a given node is referred to as its **downstream** node or node set.*

**Definition 3** (Model Decision). *Let $y \in \mathbb{R}^{\mathcal{C}}$ be the model's output over $\mathcal{C}$ classes. We define the* ***model decision*** *as the index $c^*$ such that $y^{(c^*)} > y^{(i)}$ for all $i \neq c^*$, i.e., the strict argmax.*

**Definition 4** (Causal Subpath Reference and Causal Referencing). *Given a node set $V$, we define a subpath reference as the collection of all downstream node sets that serve to bridge $V$ to the output. If every node set in this collection is causal (i.e., satisfy Definition 5), we refer to the collection as a* ***causal subpath reference***. *Thus, referencing for $V$ means considering all of its subpath references, and* ***causal referencing*** *for $V$ means considering only those that are causal.*

**Remark 1.** *Without causal referencing, one cannot guarantee that an identified node set is a true cause of the model's decision. This is because evaluating its true causal effect requires considering all possible combinations of downstream node sets (see Appendix Example 1).*

**Definition 5** (Causal Node Set). *Given a transformer interpreted as a causal graph $\mathcal{G}$, a causal subpath reference $P \subseteq \mathcal{P}$ connecting a node set $V \subseteq \mathcal{V}$ to the output, and an off-path node set $\hat{V}$ such that $P \cup \hat{V}$ equals the set of all nodes except the ancestors of $V$, and $P \cap \hat{V} = \emptyset$, we say that $V$ is* ***causal*** *for the decision if the following conditions are satisfied:*

*(5.a)* ***Necessity (Counterfactual):*** *Let $V'$ denote that $V$ is intervened on to take a different value, and let $\hat{V}'$ be defined analogously for $\hat{V}$ to causally isolate $V$. Under these interventions, the output $y'$ satisfies $\arg\max_i y'^{(i)} \neq c^*$, where $c^*$ denotes the decision without any intervention.*

*(5.b)* ***Sufficiency (Contingency):*** *Given $V$, even if nodes in $P$ are perturbed due to an intervention resulting in $\hat{V}'$, the output $y'$ satisfies $\arg\max_i y'^{(i)} = c^*$.*

*(5.c)* ***Causal Minimality:*** *$V$ is minimal; no strict subset of $V$ satisfies both (5.a) and (5.b).*

*Intervention in these conditions can also be expressed using Pearl's do-operator, i.e., $do(\cdot)$, as:*

$$\text{Necessity:} \quad \arg\max_i \mathbb{P}_i(y \mid V, \hat{V}, P) \neq \arg\max_j \mathbb{P}_j(y \mid do(V), do(\hat{V}), P),$$

$$\text{Sufficiency:} \quad \arg\max_i \mathbb{P}_i(y \mid V, \hat{V}, P) = \arg\max_j \mathbb{P}_j(y \mid V, do(\hat{V}), P),$$

*where $\mathbb{P}_i(y \mid \cdot)$ equals $y^{(i)}$, the logit corresponding to index $i$.*

**Remark 2.** *Causal candidates must be considered at the level of node sets, rather than individual nodes. This is because, when assessing the influence of parent nodes on a child node, one must account for potential interactions among those parent nodes.*

**Remark 3.** *To determine whether a node set is a cause of the decision, we must check that the decision is preserved when all other node sets are intervened away, and that it changes when the influence of that node set itself is removed. This allows us to capture both disjunctive causes (multiple independent causes) and conjunctive causes (causes that only work together). If a cause were not required to be minimal, the entire model would always be marked as causal, leading to over-attribution. This is why our framework verifies all three conditions—necessity, sufficiency, and causal minimality.*

**Definition 6** (Causal Path). *We define a* ***path*** *as a sequence of node sets connected through direct computational dependencies, i.e., edges. We also refer to a path as a* ***subpath*** *when emphasizing or isolating a specific continuous portion of a given path. Given a transformer interpreted as a causal graph $\mathcal{G}$, we define a* ***causal path*** *as a sequence of connected causal node sets.*

**Definition 7** (Reliability of Causal Evaluation). *The* ***reliability*** *of causal evaluation is defined as the proportion of considered paths to all possible decision paths. An evaluation that accounts for all possible paths (or is theoretically guaranteed to do so) has full reliability (i.e., 1).*

For a decision, there may exist multiple causal paths; we denote the set of such paths by $\mathcal{P}$. With slight abuse of notation, we use $\mathcal{P}$ to denote not only the set of sequences but also the set of all causal node sets contained in these paths, assuming some implicit ordering turns the latter into a sequence——a minor issue since the order is already determined in tracing; thus we adopt this convenience.

Having established the definitions with respect to causal paths, we present the structural Property 1 that highlights their recursive nature: under a given decision, any causal node set must have at least one parent node set that is also causal. This recursive property enables us to identify exhaustive causal paths by ensuring that causal influence can be traced backward through successive parent node sets.

**Property 1** (Causal Edge). *Within a transformer, every child node has a direct computational dependency (i.e., an edge) with its parent nodes. Under sufficient intervention (as defined in Definition 8), at least one of its parent node sets becomes a causal node set; put differently, even when no proper subset of the parent nodes is causal, the full parent node set is necessarily causal, due to the structural characteristics of transformers that satisfy the conditions of Definition 2 (see Appendix Example 2).*

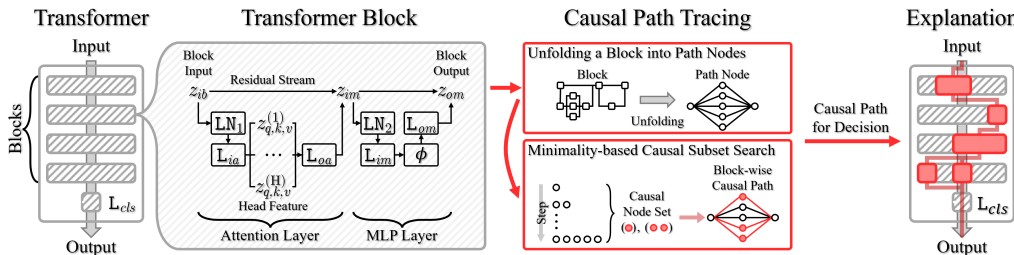

Figure 1: **Overview of our causal path tracing.** L: linear layer, LN: layer normalization, $z$: feature.

## 2.2 INTERVENTION FOR CAUSAL ISOLATION

As established in the preceding definitions, applying an intervention to isolate a node set is essential for identifying its causal influence in transformers. While there are various possible forms of intervention, it is not always clear whether they guarantee causal isolation under our setting. To address this, we formally define a *sufficient intervention* in Definition 8, to serve as a basis for assessing whether an intervention achieves reliable causal isolation in our framework.

**Definition 8** (Sufficient Intervention). *Given a transformer interpreted as a causal graph $\mathcal{G}$, we say that a node set $V$ is **sufficiently intervened** to $V'$ if the following conditions are satisfied:*

*(8.a)* ***Causal Structural Isomorphism:*** *The graphical structure of $V$ in $\mathcal{G}$, namely the adjacency structure between $V$ and its neighboring nodes, must differ from that of $V'$, and their corresponding mathematical structures (i.e., structural equations) must likewise differ. This reflects a one-to-one correspondence between graphical and mathematical structures.*

*(8.b)* ***Causal Edge Validity:*** *Given that Property 1 holds in $\mathcal{G}$, it must still hold even after an intervention on $V$. That is, the intervention must not violate the conditions specified in Definitions 2 to 6 with respect to causal paths in $\mathcal{G}$.*

*(8.c)* ***Intervention Controllability:*** *The intervention must replace a representation with one that is drawn from the model's in-distribution at the corresponding layer. In other words, although the value is forcibly assigned, it must remain on the representation manifold learned by the model, to avoid confounding effects from off-distribution artifacts or unknown factors.*

Among possible intervention strategies in transformers—such as adding noise directly to the target node (DIRECT NOISE), forwarding a noise-perturbed token embedding NOISE TOKEN), or zero-masking (ZERO MASK)—we adopt TOKEN RESAMPLING, which intervenes by replacing the target node with alternative token embeddings. The rationale for why this qualifies as a sufficient intervention is provided in Appendix Section D.

## 2.3 UNFOLDING TRANSFORMER BLOCK

In this section, we introduce a mathematical formulation of paths within a standard transformer. We begin by representing paths in a single block to establish the idea of our approach. Subsequently, we extend this formulation to cover all possible paths from the given input to the output.

Given an input $x$, our goal is to identify *which structures* within the transformer contribute to the decision as causal paths. This requires identifying the causal node sets from the decision, which in turn involves exploring the model in the backward direction. To this end, we first decompose a single block into circuits, i.e., paths, as follows (notation is provided in Figure 1):

$$
\left[ [z_q^{(h)}]_{h=1}^H;\ [z_k^{(h)}]_{h=1}^H;\ [z_v^{(h)}]_{h=1}^H \right] = \mathtt{L}_{ia}(\mathtt{LN}_1(z_{ib})),
$$
$$
z_{oa} = \mathtt{L}_{oa}([\ \mathrm{softmax}(z_q^{(h)} z_k^{(h)\top}/\sqrt{d_h}) z_v^{(h)} ]_{h=1}^H),
$$
$$
z_{im} = z_{ib} + z_{oa},
$$
$$
z_{om} = \mathtt{L}_{om}\left(\phi(\mathtt{L}_{im}(\mathtt{LN}_2(z_{im})))\right),
$$
$$
z_{ob} = z_{im} + z_{om}, \tag{1}
$$

where $z_{ib}, z_{oa}, z_{im}, z_{om}, z_{ob} \in \mathbb{R}^{T \times d_m}$ and $z_q^{(h)}, z_k^{(h)}, z_v^{(h)} \in \mathbb{R}^{T \times d_h}$, with $T$ denoting the number of tokens, $d_m$ the model dimension, and $d_h = {}^{d_m}\!/_H$. In addition, $\mathtt{L}_{ia}$ is defined with respect to its input $z$ as $\mathtt{L}_{ia}(z) = zW_{ia}^\top + b_{ia}$, and $\mathtt{L}_{oa}$, $\mathtt{L}_{im}$, and $\mathtt{L}_{om}$ are defined analogously. Here, to identify

the structures causally involved in the decision from the input, we treat the block input $z_{ib}$ as a single node. If the block output $z_{ob}$ can be unfolded with respect to $z_{ib}$, this allows us to capture the structures in a path-wise manner ($z_{ib} \sim z_{ob}$), all at once, rather than laboriously analyzing them one by one in a structure-wise fashion.

However, due to the presence of non-linear functions, i.e., softmax and GeLU $\phi$, it is nontrivial to decompose the above equations into a single unified expression. To address this, we employ a minor computational trick that rewrites the non-linear functions in the form of Hadamard products: $\text{softmax}(z/\sqrt{d_h}) = z \odot D_\alpha$ and $\phi(z) = z \odot D_\beta$, where the scaling factors $D_\alpha$ and $D_\beta$ are treated as fixed values once computed from the input $z$. In our implementation, these factors are obtained by first computing the outputs of softmax and GeLU, storing them in memory, and then performing element-wise division with $z$. In addition, we apply a similar simplification to layer normalization by treating its input-dependent statistics, mean and variance, as fixed after computation: $\text{LN}(z) = zW_{ln}^\top + b_{ln}$. Together, these interpretations allow us to express the block in a form that structurally resembles a composition of linear operations and element-wise products, as follows:

$$
z_{oa} = (\sum_{h=1}^{H} z_q^{(h)} z_k^{(h)\top} \odot D_\alpha z_v^{(h)} W_{oa}^\top) + b_{oa},
$$

$$
z_{om} = z_{oa} W_{ln_2}^\top W_{im}^\top \odot D_\beta W_{om}^\top + z_{ib} W_{ln_2}^\top W_{im}^\top D_\beta W_{om}^\top
$$
$$
+ b_{ln_2} W_{im}^\top \odot D_\beta W_{om}^\top + b_{im} \odot D_\beta W_{om}^\top + b_{om},
$$

$$
z_{ob} = z_{ib} + z_{oa} + z_{om}
$$

$$
= \underbrace{z_{ib}}_{\text{Residual Only (1 Path)}} + \underbrace{\sum_{h=1}^{H} z_q^{(h)} z_k^{(h)\top} \odot D_\alpha z_v^{(h)} W_{oa}^\top + b_{\text{attn}}}_{\text{Attention Only (H Paths)}} + \underbrace{z_{ib} W_{ln_2}^\top W_{im}^\top D_\beta W_{om}^\top + b_{\text{mlp}}}_{\text{MLP Only (1 Path)}}
$$

$$
+ \underbrace{\sum_{h=1}^{H} z_q^{(h)} z_k^{(h)\top} \odot D_\alpha z_v^{(h)} W_{oa}^\top W_{ln_2}^\top W_{im}^\top \odot D_\beta W_{om}^\top + \frac{b_{oa} W_{ln_2}^\top W_{im}^\top \odot D_\beta W_{om}^\top}{H} + b_{\text{attn+mlp}}}_{\text{Attention+MLP (H Paths)}},
$$

$$
\text{(2)}
$$

Here, $b_{\text{attn}}$, $b_{\text{mlp}}$, and $b_{\text{attn+mlp}}$ represent the terms in the block output $z_{ob}$ that do not directly involve the block input $z_{ib}$ (the full derivation is provided in Appendix Section E). By unfolding $z_{ob}$ with respect to $z_{ib}$, we obtain a set of additive terms, which can be grouped into distinct paths depending on the layer parameters involved, attention, MLP, or none. Ultimately, we can *treat each of these paths as a single node* to assess its causal contribution.

Note that we omit the unfolding of $z_q^{(h)}, z_k^{(h)}$, and $z_v^{(h)}$ for brevity, as they are linear functions of $z_{ib}$ via $W_{ln_1}, W_{ia}$ and follow the same path structure. Furthermore, for bias terms, although in principle each attention head could receive a different degree of contribution from the bias, for the sake of tractable analysis we assume that they contribute equally to all head-specific paths.

## 2.4 Minimality-based Causal Subset Search per Block

As shown earlier, all paths within a block can be decomposed into additive terms, each treated as an individual node. Based on this, we perform a block-wise backward search for causal node sets to trace the causal path for a given decision. Here, since path-level interactions must be considered, all possible combinations of path nodes within each block need to be evaluated. However, a brute-force approach incurs a complexity of $O(2^n)$, as it requires enumerating all subsets, making exhaustive search impractical for large-scale models. Although such exponential growth is unavoidable in the worst case, we propose a strategy based on Condition (5.c) that enables polynomial-time search on average.

The core idea, based on Condition (5.c), is that a causal node set must be minimal. That is, if a subset $V \subseteq V_p$ is identified as a causal node set, where $V_p$ denotes the set of all path nodes within a block, then any superset of $V$ cannot be minimal and thus does not need to be evaluated. Building on this, our search strategy proceeds in steps by subset size, starting from the smallest. As illustrated in Algorithm 1, causal node sets identified at smaller steps are used to prune the search space at larger steps by eliminating supersets that violate minimality. This strategy leads to an average-case time complexity that is polynomial in practice, as formally analyzed in Theorem 1 (proof in Appendix).

---

**Algorithm 1** Minimality-based Causal Subset Search per Block

---

1: **Input:** A path node set $V_p = [v_1, \ldots, v_n]$ from a specific block (i.e., additive terms within the block), a subgraph $\mathcal{G}_c$ (downstream blocks of $V_p$ in the transformer $\mathcal{G}$), causal subpaths $P$ connecting $V_p$ to the decision $c^*$ in the output $y$, and an off-path node set $\hat{V}$ such that $P \cup \hat{V}$ equals the set of all nodes except the ancestors of $V$, and $P \cap \hat{V} = \emptyset$
2: **Output:** $V_{\text{out}} = \{V_1, V_2, \ldots\}$, where each $V_i \subseteq V_p$ satisfies Conditions (5.a), (5.b), and (5.c)
3: $V_{\text{out}} \leftarrow \emptyset$
4: **for** $s = 1$ to $n$ **do**            ▷ Subset size
5:     **for** each $V \subseteq V_p$ such that $|V| = s$ **do**
6:         **if** $V_i \subseteq V$ for some $V_i \in V_{\text{out}}$ **then**
7:             **continue**            ▷ Fail Condition (5.c) (causal minimality)
8:         **end if**
9:         Intervene on $V \rightarrow V'$, and on $\hat{V} \rightarrow \hat{V}'$
10:         Let $y' \leftarrow$ model output under $(V', \hat{V}', P)$          ▷ for Condition (5.a) (necessity)
11:         Let $y'' \leftarrow$ model output under $(V, \hat{V}', P)$          ▷ for Condition (5.b) (sufficiency)
12:         **if** $\arg\max_i y'^{(i)} = c^*$ **or** $\arg\max_i y''^{(i)} \neq c^*$ **then**
13:             **continue**            ▷ Fail Condition (5.a) or (5.b)
14:         **end if**
15:         $V_{\text{out}} \leftarrow V_{\text{out}} \cup \{V\}$            ▷ Satisfies Conditions (5.a), (5.b), (5.c)
16:     **end for**
17: **end for**
18: **return** $V_{\text{out}}$

---

**Theorem 1** (Expected Time Complexity of Minimality-based Subset Search). *Consider a minimality-based subset search over $n$ nodes, where each subset is independently selected as a causal node set with probability $p$. Then, the expected number of subset evaluations over all subsets is bounded by:*

$$n + (1-p) \times \sum_{s=2}^{n} \max \left( 0, \binom{n}{s} + \sum_{i=1}^{s-1} \sum_{m=1}^{\lfloor p\binom{n}{i} \rfloor} (-1)^m \binom{p\binom{n}{i}}{m} \binom{n-mi}{s-mi} \right). \tag{3}$$

*Given this, the expected time complexity grows approximately as:*

$$O\left( n^{\lfloor \log_2 \left( \frac{1}{p} + 2 \right) \rfloor} \right). \tag{4}$$

**Remark 4.** *Although the exact value of $p$ is unknown, the time complexity, depending on $p$, is polynomial in the best and average cases. For example, when $p = 1$, all subsets of size $s \geq 2$ are pruned, so only singleton subsets are evaluated, resulting in a time complexity of $O(n)$. However, since the underlying search space contains $2^n$ possible subsets, exponential complexity is unavoidable in the worst case. Nonetheless, such worst-case scenarios occur only infrequently; for example, when $p \leq \frac{1}{2^n - 2}$, causal node sets are rarely selected at each step, requiring exhaustive search over all subset combinations and leading to a time complexity of $O(2^n)$.*

## 2.5 Unfolded Block-wise Causal Path Tracing

In this section, we extend the minimality-based causal subset search from a single block to the entire transformer. We traverse blocks backward, identifying causal node sets and updating the causal path reference $P$ at each step. Using each causal set individually as $P$ is computationally expensive, as it requires repeated searches. Instead, we use their union as the reference, which significantly reduces the cost. As in Theorem 2, the union-based strategy ensures that reliability converges to 1 (proof in Appendix), indicating near-complete causal coverage. Algorithm 2 outlines the full procedure.

Note that, as mentioned earlier (below Definition 7), the processes in lines 8 and 9 are expressed, for convenience, as a set of node sets. By Theorem 2, however, all combinations of causal node sets from adjacent blocks within $P$ and $\mathcal{P}$ constitute causal paths; thus, it can also be implicitly regarded as a set of sequences.

**Theorem 2** (Causal Union Reference Reliability). *Consider a minimality-based subset search over $n$ nodes, where each subset is independently selected as a causal node set with probability $p$. Suppose that a collection of such sets, $V_{out}^{(j+1)} = \{V_i^{(j+1)}\}_{i=1}^{k}$, is identified as causal in the $(j+1)$-th block. Their union, denoted as $P = \bigcup_{i=1}^{k} V_i^{(j+1)}$, serves as the causal subpath reference for the minimality-*

---

**Algorithm 2** Unfolded Block-wise Causal Path Tracing

---

1: **Input:** A transformer $\mathcal{G}$ with $D$ blocks, and a model output $y$ with decision $c^*$ for a given input
2: **Output:** Causal paths $\mathcal{P}$ composed of the causal node sets identified in every block
3: $P \leftarrow \{\mathtt{L}_{cls}\}$  $\triangleright$ By Property 1, the classifier $\mathtt{L}_{cls}$ serves as the initial causal path reference
4: $\mathcal{P} \leftarrow \emptyset; \quad \mathcal{G}_c \leftarrow \{\mathtt{L}_{cls}\}$
5: **for** $j = D$ to 1 **do**  $\triangleright$ Iterate backward through transformer blocks
6: $\quad$ Let $V_p^{(j)} \leftarrow$ unfolded path nodes in block $j$
7: $\quad V_{\text{out}}^{(j)} \leftarrow \text{MIN\_SEARCH}(V_p^{(j)}, \mathcal{G}_c, P, c^*)$  $\triangleright$ See Algorithm 1
8: $\quad P \leftarrow \bigcup V_{\text{out}}^{(j)}$  $\triangleright$ Update causal path reference (see Theorem 2)
9: $\quad \mathcal{P} \leftarrow \mathcal{P} \cup V_{\text{out}}^{(j)}; \quad \mathcal{G}_c \leftarrow$ block $j$
10: **end for**
11: **return** $\mathcal{P}$

---

based subset search in the $j$-th block. Let $s_{avg}$ denote the average size of the $k$ causal node sets in $V_{out}^{(j+1)}$. Then, the reliability (see Definition 7) of the resulting causal node set obtained using $P$ is given by:

$$p + (1 - p) \left( 1 - \left( 1 - \frac{s_{avg}}{n} \right)^k \right)^n \to 1 \tag{5}$$

## 3 EXPERIMENTS

### 3.1 MODELS, DATASETS, AND BASELINES

We conduct experiments on five transformer models: three language models (GPT2-xs (Algorithmic Research Group, 2025), Pythia-14m and Pythia-1b (Biderman et al., 2023)) and two vision models (ViT-tiny (Dosovitskiy et al., 2020) and DeiT-tiny (Touvron et al., 2021)). For language tasks, we use the KNOWNS1000 (Meng et al., 2022a) and T-REX (Elsahar et al., 2018; Petroni et al., 2019) datasets. For vision tasks, we evaluate on IMAGENET (Russakovsky et al., 2015) and OFFICEHOME (Venkateswara et al., 2017). Further results are provided in Appendix.

As summarized in Table 1, we compare against existing methods that are feasible for decision-level path tracing. To enable fair comparison with our method, all baselines are extended under a backward chaining framework, assuming that residual connections are always present.

Specifically, $\text{NT}_1$ and $\text{NT}_{10\%}$ are adaptations of the node-level patching method by Meng et al. (2022a), referred to as Node-level patching-based Tracing (NT), where the top-1 node ($\text{NT}_1$) or the top 10% of nodes ($\text{NT}_{10\%}$), ranked by their estimated effect within each block, are selected as decision paths. $\text{ET}_{\text{all}}$ and $\text{ET}_{\text{cls}}$ are based on the edge-level patching method by Syed et al. (2024), referred to as Edge-level patching-based Tracing (ET), which assumes task-level edge attribution. Here, a "task" is defined as either the entire dataset ($\text{ET}_{\text{all}}$) or a single class ($\text{ET}_{\text{cls}}$). ET-IG$_{\text{all}}$ and ET-IG$_{\text{cls}}$ are based on another edge-level patching method by Hanna et al. (2024), which enhances faithfulness by leveraging integrated gradients (Sundararajan et al., 2017) from $\text{ET}_{\text{all}}$ and $\text{ET}_{\text{cls}}$.

Note that path-level patching methods are not included in the comparison, as no existing method feasibly enumerates all decision paths for a given output—our method is the first to make this feasible. We refer to our approach as Causal Path Tracing (CPT). Implementation details are in Appendix.

### 3.2 RESULTS

**Minimality-based search converges empirically in polynomial time; furthermore, it reveals how models rely on path-level reasoning** Figure 2 presents the empirical time complexity of our causal path tracing procedure across models. Each subplot shows the distribution of $s^*$, defined as the final step in the minimality-based search where no further superset remains due to pruning by already selected causal node sets. The average $s^*$ is used to estimate the empirical probability $p$ that a randomly selected subset is causal (see Theorem 1), which is then compared against the theoretical lower bound required for polynomial-time search (see Remark 4).

In all models, the empirical $p$ exceeds the theoretical threshold, confirming that the proposed search converges in polynomial time in practice, as predicted. Notably, the search typically completes in few steps, with pruning often concluding well before the midpoint of the search space.

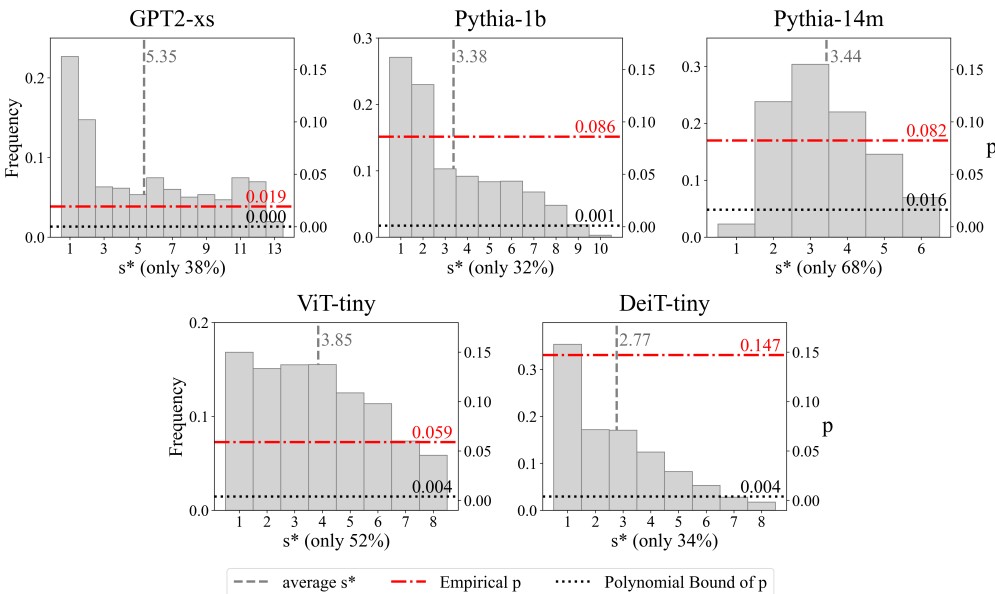

Figure 2: **Comparison of empirical time complexity.** Causal path tracing under our method runs in polynomial time across models. Each subplot shows the reduced search space (in parentheses); $s^*$ is the maximum step reached by the minimality-based search (i.e., the largest $s$ such that the term in Equation (3) is nonzero); the empirical $p$ estimated from the average $s^*$ (see Theorem 1); and the polynomial bound of $p$, which is the theoretical lower bound required to ensure polynomial-time search (see Remark 4). Language models use T-Rex; vision models use ImageNet.

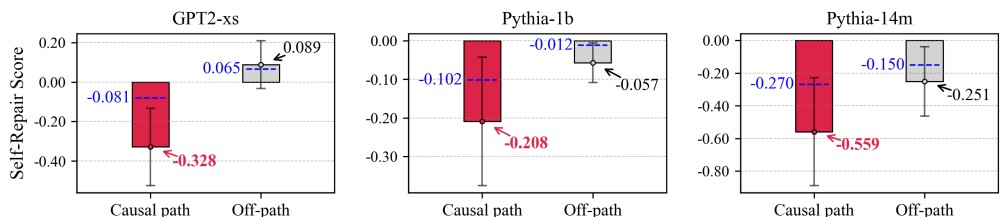

Figure 3: **Self-repair scores on causal path vs. off-path components.** Each bar shows the mean (dot with arrow) and standard deviation (error bar); medians are shown as blue dashed lines. Lower scores indicate less self-repair. Results are averaged over KNOWNS1000 and T-REX.

The distribution of $s^*$ also reveals how the model leverages internal structure for decision making: a small $s^*$, especially when concentrated near one, indicates that the model relies primarily on the strength of individual paths; in contrast, a larger or more dispersed $s^*$ suggests that reasoning involves interactions among multiple paths rather than relying on any single strong one.

**Causal path components exhibit lower self-repair, suggesting irreplaceable decision signals** We compare self-repair scores between attention heads on the causal path and those off the path, as identified by our tracing method. Following Rushing & Nanda (2024), we categorize components based on whether they belong to the traced causal path and measure their self-repair accordingly.

As shown in Figure 3, we find that self-repair occurs less frequently on the causal path. While self-repair is known to be noisy, as noted by Rushing & Nanda (2024), the results show a difference: both the mean and median scores are consistently lower on the causal path than off it. This suggests that the causal path captures components essential to the decision and less reliant on backup mechanisms. In other words, the selected paths carry information not easily replaceable, underscoring their critical role for decision (see Appendix Sections G.2 to G.4 for further details).

**Class-specific causal subpaths play a functional role in predicting their respective classes** Here, we aim to investigate whether the discovered causal paths contain class-wise causal nodes—nodes that

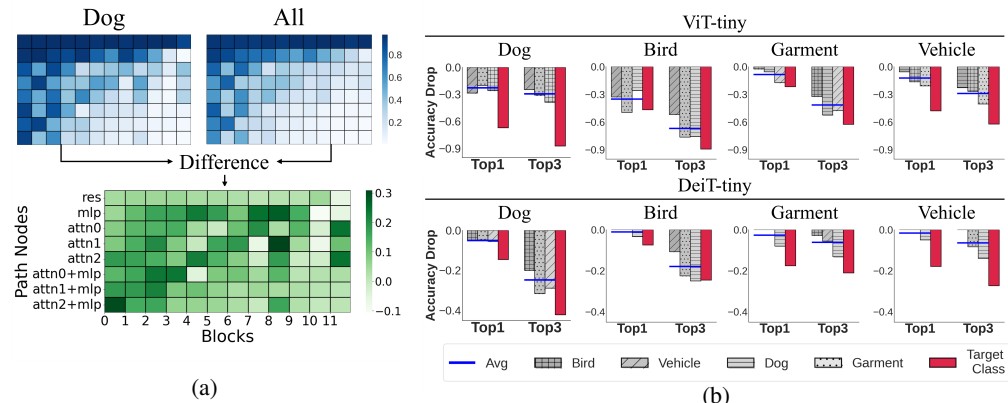

Figure 4: **Causal paths uniquely activated for specific classes. (a)** Average causal path ratios for a target class (left), all classes (right), and their difference (bottom), highlighting class-specific paths. Here, res, mlp, and attn# indicate residual, MLP, and attention paths from head #, respectively. **(b)** Accuracy drop when ablating the most class-specific path, showing selective reliance by each class.

|  | Hit. ($\uparrow$) | Faith. ($\uparrow$) | Spars. ($\downarrow$) |
|---|---|---|---|
| $NT_1$ | 0.0000 | 0.0005 | 0.6571 |
| $NT_{10\%}$ | 0.0000 | 0.0006 | 0.5648 |
| $ET_{all}$ | 0.2079 | 0.2354 | 0.9806 |
| $ET_{cls}$ | 0.4808 | 0.4734 | 0.9909 |
| $ET\text{-}IG_{all}$ | 0.6677 | 0.7496 | 0.9938 |
| $ET\text{-}IG_{cls}$ | 0.7351 | 0.7490 | 0.9923 |
| **CPT** | **0.9826** | **0.5466** | **0.8641** |

Table 2: **Quantitative results (language).** Averaged over three models on two datasets.

|  | Hit. ($\uparrow$) | Faith. ($\uparrow$) | Spars. ($\downarrow$) |
|---|---|---|---|
| $NT_1$ | 0.0105 | 0.0136 | 0.7276 |
| $NT_{10\%}$ | 0.0078 | 0.0133 | 0.0799 |
| $ET_{all}$ | 0.4454 | 0.3166 | 0.9999 |
| $ET_{cls}$ | 0.2627 | 0.1832 | 0.9650 |
| $ET\text{-}IG_{all}$ | 0.2408 | 0.2312 | 0.9817 |
| $ET\text{-}IG_{cls}$ | 0.3002 | 0.2000 | 0.9499 |
| **CPT** | **0.9638** | **0.2991** | **0.7280** |

Table 3: **Quantitative results (vision).** Averaged over two models on two datasets.

are consistently utilized across samples within the same class group—and whether these nodes play a significant role in the model's classification decisions. To improve clarity, we first select four super-classes—dog, bird, garment, and vehicle—among the 1,000 ImageNet classes based on semantic similarity derived from WordNet. We then aggregated the causal paths extracted from individual samples and compiled statistics on the frequency of each subpaths' occurrence. By comparing these frequencies to the overall average across all samples, we identified causal subpaths that were significantly more active within specific super-classes (as shown in Figure 4-(a)). We refer to these as class-wise causal subpaths, hypothesizing that *they store key discriminative information relevant to their respective super-classes* due to their unusually high activation rates.

To validate this hypothesis, we intervene in the class-wise causal subpaths and measure the performance drop. If these nodes indeed encode class-specific information, their removal should lead to a greater accuracy drop within the corresponding super-class than in others. Figure 4-(b) clearly demonstrates this pattern. For instance, when the class-wise causal subpaths for the dog super-class were deactivated in a ViT-tiny model, the top-1 accuracy for dog samples decreased by approximately $44.7\%$ more than that for other super-classes. Similar trends were observed across bird, garment, and vehicle classes, indicating that the proposed metric functions consistently across the model.

It is important to note that due to inherent semantic overlap among ImageNet classes, interventions on class-wise causal subpaths may still affect the logits of unrelated classes. Additionally, due to visual diversity within each super-class, turning off only a small number of subpaths may not entirely collapse performance. Nevertheless, the consistent and pronounced patterns observed across all super-classes suggest that our method effectively identifies causal subpaths that play a meaningful role in class-specific inference (see Appendix Section G.5 for further details).

**Quantitative results show our method yields reliable and faithful explanations**  Each value in Tables 2 and 3 represents the average score across models on two datasets. All methods are evaluated

by pruning the model to retain only the paths identified by each method. We report three metrics: **Hit.** (hit rate) score measures the proportion of cases in which the pruned model produces the same decision as the original; **Faith.** (faithfulness) score quantifies the ratio of the original logit preserved after pruning; and **Spars.** (sparsity) score denotes the proportion of model parameters retained by the identified path.

Our method (CPT) achieves a near-perfect hit rate score, consistent with the theoretical guarantee in Theorem 2 that the identified paths are reliably causal. In contrast, existing methods show substantially lower hit rate scores, supporting our claim in Table 1 that while tracing is feasible with backward chaining, it is generally not reliable for identifying true decision paths.

With respect to faithfulness score, CPT also shows a relatively high score, but ET-IG achieves even higher one. However, the fact that its hit rate is not close to 1 implies that, when retaining only the identified path, the model often fails to reproduce the original decision. In this context, a high faithfulness score merely indicates that the logit for the original decision is partially preserved, but not necessarily dominant; i.e., other logits may become even larger in ET-IG, *leading to a different prediction*. Therefore, by jointly considering the hit rate and faithfulness scores, our method is the most faithful in explaining the given decision. Notably, it does so while retaining significantly fewer parameters: whereas edge-level methods such as ET and ET-IG rely on nearly the entire model, CPT yields more compact explanations through efficient path selection (see Appendix Section G.6).

## 4 CONCLUSION

In this paper, we presented an automated framework for tracing causal paths given a decision. We provide both theoretical analysis and empirical evidence showing that our method efficiently uncovers all causal paths responsible for a decision, with average-case polynomial-time complexity. Furthermore, we demonstrated that the identified causal paths (1) are less susceptible to self-repair effects, (2) reveal the structural grounds for subpaths uniquely activated for specific classes, and (3) yield more faithful and precise explanations than existing methods.

**Limitations and Future Work.** First, the identified causal paths are derived under the assumptions of our proposed framework and may not generalize under different assumptions. In particular, our unfolding procedure assumes uniform propagation of bias terms across all paths and a pre-computation trick for non-linear operations; however, precisely quantifying their individual contributions is non-trivial and remains an open problem for future work. Second, we acknowledge that our experiments were conducted on smaller models compared to state-of-the-art architectures. Large models may still incur prohibitive runtime in worst-case scenarios, and the reduced search space can remain sizable. Nevertheless, it is important to note that even on smaller models other methods could not cover the total search space (see Appendix G.7), whereas our method can do so with polynomial-time complexity on average. Extending our minimality-based subset search to also prune supersets of non-causal subsets could mitigate this issue. Lastly, while our analysis focuses on structural mechanisms within the model, it opens avenues for future integration with feature attribution methods, potentially bridging structural and feature-level interpretability.

Despite these limitations, our work is the first to propose an efficient and reliable framework for tracing causal paths within transformer models for a given decision. We believe this represents an important step toward making transformers more transparent and robust in safety-critical domains, helping to prevent misuse and improve trust in deployment.

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

## A  DECLARATION OF LLM USAGE

We used an LLM solely for grammar correction.

## B  RELATED WORK

Mechanistic interpretability aims to identify how internal components—such as attention heads and MLPs—contribute to model behavior. Early work explored circuit discovery in simplified settings (Elhage et al., 2021), while recent methods adopt Pearl's causal theory (Pearl, 2009) and ablation-based interventions to reveal components essential to specific outputs.

**Node-level patching**   These methods aim to identify the contribution of individual input features to a model's prediction. Vig et al. (2020) investigate gender bias through causal mediation analysis by selectively patching specific input features. Wang et al. (2022) ablate features one by one to study their roles in indirect object identification, and further report the phenomenon of backup behavior, where other components compensate for ablated information. Meng et al. (2022a) restore specific token-level features to identify where factual knowledge is stored in the model when predicting objects in sentences, and demonstrate direct editing of such representations. Building on this, Meng et al. (2022b) extend the editing framework to perform large-scale knowledge editing across many memory locations. Heimersheim & Janiak (2023) analyze individual input features to uncover the mechanisms behind repeated argument names in Python docstring generation. Zhang & Nanda (2024) systematically examine the impact of various methodological choices in node-level patching, providing a comprehensive evaluation of this line of research.

**Edge-level patching**   These methods examine the influence of neighboring feature pairs with direct computational dependencies. Conmy et al. (2023) propose a greedy edge ablation algorithm to automatically abstract model behavior by iteratively removing dependencies and observing the impact on predictions. Syed et al. (2024) accelerate edge-level analysis by introducing a gradient-based first-order approximation that estimates the effect of ablating each edge efficiently. Hanna et al. (2024) enhances faithfulness by leveraging integrated gradients (Sundararajan et al., 2017). Bhaskar et al. (2024) replace missing edges with counterfactual activations obtained from corrupted examples, enabling gradient-based pruning without relying on discrete search or linear approximations.

**Path-level patching**   These approaches investigate the contribution of distant feature pairs connected through multiple accumulated dependencies. Chan et al. (2022) justify that ablations using zero or mean activations can induce misleading interventions, and instead propose hypothetical tests to explore decision paths. Hanna et al. (2023) manually specify and patch internal paths in a language model to explain the circuit responsible for a year-span prediction task. Nanda et al. (2023) manually ablate specific paths to investigate where factual knowledge related to sports is stored within the model.

When applying existing methods to trace paths for a given decision, node- and edge-level approaches are feasible under a backward chaining framework. However, they lack reliability as they perform chaining without causal referencing, making it difficult to guarantee the correctness of the identified paths. Path-level methods, on the other hand, often rely on hypothetical tests or require manual specification of paths, which renders full explanations infeasible due to the combinatorial complexity of possible paths. In contrast, our method enables reliable identification of causal paths while remaining computationally efficient, achieving tractable search on average in polynomial time.

## C  SUPPLEMENTARY DEFINITIONS

We restate Definition 4 and Remark 1 here, now including an example for clarity.

**Definition 4** (Causal Referencing). *Given a node set $V$, we define a subpath reference as the collection of all downstream node sets that serve to bridge $V$ to the output. If every node set in this collection is causal (i.e., satisfy Definition 5), we refer to the collection as a **causal subpath reference**. Thus, referencing for $V$ means considering all of its subpath references, and **causal referencing** for $V$ means considering only those that are causal.*

**Remark 1.** *Without causal referencing, one cannot guarantee that an identified node set is a true cause of the model's decision. This is because evaluating its true causal effect requires considering all possible combinations of downstream node sets.*

**Example 1.** *Consider a two-block transformer in which two node sets $V_1^{(2)}$ and $V_2^{(2)}$ in the final block have already been identified as causal node sets, respectively. In this case, whether a node set $V_1^{(1)}$ in the first block is causal may depend on whether it is evaluated jointly with $V_1^{(2)}$ and $V_2^{(2)}$, or both (the case of neither can be safely ignored due to Property 1). That is, the causal status of $V_1^{(1)}$ can vary depending on which downstream path the internal information propagates through; in other words, some combinations may be sufficient for the decision, while others may not.*

Furthermore, we restate Property 1 here, now including an example for clarity.

**Property 1** (Causal Edge). *Within a transformer, every child node has a direct computational dependency (i.e., an edge) with its parent nodes. Under sufficient intervention (as defined in Definition 8), at least one of its parent node sets becomes a causal node set; put differently, even when no proper subset of the parent nodes is causal, the full parent node set is necessarily causal, due to the structural characteristics of transformers that satisfy the conditions of Definition 2.*

**Example 2.** *Consider a transformer interpreted as $\mathcal{G}$, containing only a node $v_c$ and the set $V_p$ of all parent nodes of $v_c$. Suppose that $v_c$ belongs to a causal node set (i.e., the causal node set consists of $v_c$ alone). Since $v_c$ depends on $V_p$, intervening on all of $V_p$ leads to a different decision, satisfying Condition (5.a). Moreover, if $V_p$ is unchanged, the decision remains the same since there is no off-path node set (i.e., $\hat{V} = \emptyset$), thereby satisfying Condition (5.b). Thus, $V_p$ satisfies both conditions in Definition 5, implying that it is, at least, a causal node set.*

## D   WHY TOKEN RESAMPLING QUALIFIES AS A SUFFICIENT INTERVENTION

Among possible intervention strategies within transformers, a straightforward approach is to add noise directly to the target node (DIRECT NOISE). However, such perturbation fails to satisfy Condition (8.a). Another commonly used strategy, as in prior works such as (Meng et al., 2022a), involves forwarding a noise-perturbed token embedding through the model to obtain a corrupted version of the target node (NOISE TOKEN); however, this violates Condition (8.c). A naive alternative, such as zero-masking (ZERO MASK), also proves inadequate, as it breaks Condition (8.b) by distorting Property 1.

Since these strategies fail to satisfy the required conditions in our setting, we instead adopt a method that intervenes on the target node by resampling from alternative token embeddings (TOKEN RESAMPLING). This approach satisfies all three conditions for a sufficient intervention.

To justify why TOKEN RESAMPLING, unlike other intervention methods, constitutes a sufficient intervention in our framework, we consider a simplified transformer architecture with a single block and a single attention head; thus, the block input $z_{ib}$ corresponds to the original token embedding $z_{token}$. The block output can then be decomposed into four additive path terms, as described in Equation (2): the residual only path node ($v_1$), the Attention only path node ($v_2$), Attention+MLP path node ($v_3$), and MLP only path node ($v_4$).

### D.1   FOR DEFINITION (8.A) (CAUSAL STRUCTURAL ISOMORPHISM)

Consider two intervention scenarios where either $v_1$ or $v_4$ is replaced.

Under TOKEN RESAMPLING, we substitute the original token embedding $z_{token}$ with a different in-vocabulary embedding $z'_{token}$. This substitution affects both $v_1$ and $v_4$, as they both contain $z_{token}$ (i.e., $z_{ib}$) as their root variable. Crucially, whether $z_{token}$ or $z'_{token}$ appears in $v_1$ or $v_4$ leads to distinct expressions for the block output. By these expressions, the resulting interventional graphs remain distinguishable across scenarios. This one-to-one correspondence between the mathematical form and the graph structure holds not only in this toy example but also more generally, thereby satisfying Definition (8.a).

In contrast, DIRECT NOISE adds the same Gaussian noise directly to multiple nodes (e.g., $v_1$ or $v_4$), which leads to overlapping outcomes across different intervention scenarios. This is due to

the additive nature of the block output, where all path node contributions are combined through addition. As a result, this destroys the one-to-one mapping between interventional graphs and their mathematical representations, thereby violating Definition (8.a).

## D.2    FOR DEFINITION (8.B) (CAUSAL EDGE VALIDITY)

Consider an intervention scenario where all path nodes are replaced.

Under ZERO MASKING, each component is zeroed out, which causes the model output to collapse to a constant vector (i.e., the block output $z_{ob} = 0$), resulting in a fixed decision $y = b_{cls}$, where $b_{cls}$ denotes the classifier's bias term. In this case, the intervened decision becomes entirely independent of the original input, relying solely on the classifier bias. As a result, there exists at least one decision for which the causal necessity condition is never satisfied, thereby violating Definition (8.b).

In contrast, TOKEN RESAMPLING prevents the decision from collapsing to the classifier bias $b_{cls}$ by replacing the original token embedding $z_{token}$ with a different in-vocabulary embedding $z'_{token}$, ensuring that the causal necessity condition can still be satisfied and thus upholding Definition (8.b).

## D.3    FOR DEFINITION (8.C) (INTERVENTION CONTROLLABILITY)

Consider an intervention scenario where $v_4$ is replaced.

TOKEN RESAMPLING replaces a representation with another token embedding from the model's learned vocabulary. Because these embeddings originate from the model itself, the resulting representations remain on the model's manifold, ensuring that interventions stay within the model's natural distribution and thus satisfying Definition (8.c).

In contrast, NOISE TOKEN replaces $z_{token}$ with $z_{token} + z_{noise}$, where $z_{noise}$ is the noise. As this composite input is passed through the MLP in $v_4$, the noise term is transformed by learned parameters, producing an activation that is not guaranteed to lie on the model's representation manifold. Furthermore, as this noisy signal propagates through subsequent layers, this may lead to out-of-distribution representations, thereby violating Definition (8.c).

# E  FULL DERIVATION OF UNFOLDING TRANSFORMER BLOCK

We begin with the following equation, Equation (1):

$$
\left[ [z_q^{(h)}]_{h=1}^H;\ [z_k^{(h)}]_{h=1}^H;\ [z_v^{(h)}]_{h=1}^H \right] = \mathsf{L}_{ia}(\mathsf{LN}_1(z_{ib})),
$$
$$
z_{oa} = \mathsf{L}_{oa}([\ \mathrm{softmax}(z_q^{(h)} z_k^{(h)\top}/\sqrt{d_h}) z_v^{(h)}\ ]_{h=1}^H),
$$
$$
z_{im} = z_{ib} + z_{oa},
$$
$$
z_{om} = \mathsf{L}_{om}\left(\phi(\mathsf{L}_{im}(\mathsf{LN}_2(z_{im})))\right),
$$
$$
z_{ob} = z_{im} + z_{om}, \tag{6}
$$

where $z_{ib}, z_{oa}, z_{im}, z_{om}, z_{ob} \in \mathbb{R}^{T \times d_m}$ and $z_q^{(h)}, z_k^{(h)}, z_v^{(h)} \in \mathbb{R}^{T \times d_h}$, where $T$ denotes the number of tokens, $d_m$ the model dimension, $H$ the number of heads, and $d_h = d_m/H$. Here, this can be unfolded into $2H + 2$ paths, as follows:

$$
z_{ob} = z_{im} + z_{om} = \underbrace{z_{ib}}_{\text{Residual Only ``1 } Path\text{''}} + z_{oa} + z_{om}.
$$

$$
z_{oa} = \underbrace{(\sum_{h=1}^H z_q^{(h)} z_k^{(h)\top} \odot D_\alpha z_v^{(h)} W_{oa}^\top)}_{\substack{\text{Attention Only} \\ \text{(i.e., Attention } params. \text{ with the block input } z_{ib}, \\ \text{as contained in } z_q^{(h)}, z_k^{(h)}, z_v^{(h)})}} + \underbrace{b_{oa}}_{\substack{\text{Bias Terms from Attention} = H \cdot b_{\text{attn}} \\ \text{(i.e., Attention } params. \text{ only)}}},
$$

$$
= \underbrace{(\sum_{h=1}^H z_q^{(h)} z_k^{(h)\top} \odot D_\alpha z_v^{(h)} W_{oa}^\top + b_{\text{attn}})}_{\text{Attention Only ``}H\text{ } Paths\text{''}}.
$$

$$
z_{om} = \underbrace{z_{oa} W_{ln_2}^\top W_{im}^\top \odot D_\beta W_{om}^\top}_{\substack{\text{Attention+MLP} \\ \text{(i.e., MLP } params. \text{ with the attention output } z_{oa})}}
$$
$$
+ \underbrace{z_{ib} W_{ln_2}^\top W_{im}^\top D_\beta W_{om}^\top}_{\substack{\text{MLP Only} \\ \text{(i.e., MLP } params. \text{ with the block input } z_{ib})}} + \underbrace{b_{ln_2} W_{im}^\top \odot D_\beta W_{om}^\top + b_{im} \odot D_\beta W_{om}^\top + b_{om}}_{\substack{\text{Bias Terms from MLP} = H \cdot b_{\text{attn+mlp}} + b_{\text{mlp}} \\ \text{(i.e., MLP } params. \text{ only)}}},
$$

$$
= \left( (\sum_{h=1}^H z_q^{(h)} z_k^{(h)\top} \odot D_\alpha z_v^{(h)} W_{oa}^\top) + b_{oa} \right) W_{ln_2}^\top W_{im}^\top \odot D_\beta W_{om}^\top
$$
$$
+ z_{ib} W_{ln_2}^\top W_{im}^\top D_\beta W_{om}^\top + H \cdot b_{\text{attn+mlp}} + b_{\text{mlp}},
$$

$$
= (\sum_{h=1}^H z_q^{(h)} z_k^{(h)\top} \odot D_\alpha z_v^{(h)} W_{oa}^\top W_{ln_2}^\top W_{im}^\top \odot D_\beta W_{om}^\top) + b_{oa} W_{ln_2}^\top W_{im}^\top \odot D_\beta W_{om}^\top
$$
$$
+ z_{ib} W_{ln_2}^\top W_{im}^\top D_\beta W_{om}^\top + H \cdot b_{\text{attn+mlp}} + b_{\text{mlp}},
$$

$$
= \underbrace{(\sum_{h=1}^H z_q^{(h)} z_k^{(h)\top} \odot D_\alpha z_v^{(h)} W_{oa}^\top W_{ln_2}^\top W_{im}^\top \odot D_\beta W_{om}^\top + \frac{b_{oa} W_{ln_2}^\top W_{im}^\top \odot D_\beta W_{om}^\top}{H} + b_{\text{attn+mlp}})}_{\text{Attention+MLP ``}H\text{ } Paths\text{''}}
$$
$$
+ \underbrace{z_{ib} W_{ln_2}^\top W_{im}^\top D_\beta W_{om}^\top + b_{\text{mlp}}}_{\text{MLP Only ``1 } Path\text{''}}. \tag{7}
$$

# F  PROOFS

## F.1  PROOFS OF THEOREM 1

**Lemma 1** (Expected Evaluation Number for Theorem 1). *Consider a minimality-based subset search over $n$ nodes, where each subset is independently selected as a causal node set with probability $p$. Then, the expected number of subset evaluations over all subsets is bounded by:*

$$n + (1 - p) \times \sum_{s=2}^{n} \max\left(0, \binom{n}{s} + \sum_{i=1}^{s-1} \sum_{m=1}^{\lfloor p\binom{n}{i}\rfloor} (-1)^m \binom{p\binom{n}{i}}{m} \binom{n - mi}{s - mi}\right).$$

*Proof.* At step $s = 1$, there are $\binom{n}{1} = n$ singleton subsets, all of which must be evaluated, resulting in exactly $n$ evaluations.

For each step $s \geq 2$, the number of candidate subsets of size $s$ is $\binom{n}{s}$. A subset of size $s$ is pruned if it contains any previously selected subset of size $i < s$. Thus, the expected number of subsets to be evaluated at step $s$ is the total number of candidates minus the expected number of pruned subsets:

$$\binom{n}{s} - \mathbb{E}[\text{number of pruned subsets at step } s].$$

To estimate the pruning term, note that the number of subsets of size $i$ is $\binom{n}{i}$, and each is independently selected with probability $p$. For any such subset of size $i < s$, the number of supersets of size $s$ that include it is $\binom{n-i}{s-i}$, as the remaining $s - i$ nodes must be selected from the $n - i$ nodes not in the subset. Therefore, each selected subset of size $i$ contributes in expectation $p \cdot \binom{n-i}{s-i}$ pruned subsets at step $s$. Multiplying by the total number $\binom{n}{i}$ of such subsets yields the expected number of pruned subsets due to size-$i$ selections:

$$p\binom{n}{i}\binom{n - i}{s - i}.$$

Summing over all $1 \leq i < s$, the total expected number of pruned subsets at step $s$ is:

$$\sum_{i=1}^{s-1} p\binom{n}{i}\binom{n - i}{s - i}. \tag{8}$$

Equation (8) assumes that pruning effects from different subsets are disjoint. However, a subset of size $s$ may be pruned by multiple selected subsets of size $i$, which results in overcounting. To correct for this, we apply the inclusion–exclusion principle. Equation (8) corresponds to the first-order term of this expansion, and is fully incorporated into the more general form below.

For instance, when two subsets of size $i$ are both selected and jointly included in a size-$s$ subset, that subset is overcounted in the previous estimate and must be subtracted once. The number of such expected double-overlaps is:

$$-\sum_{i=1}^{s-1} \binom{p\binom{n}{i}}{2}\binom{n - 2i}{s - 2i}.$$

This logic generalizes to arbitrary overlap levels. When $m$ mutually disjoint subsets of size $i$ are selected and jointly included in a size-$s$ subset, the corresponding correction term alternates in sign according to the inclusion–exclusion principle, and takes the form:

$$(-1)^m \binom{p\binom{n}{i}}{m}\binom{n - mi}{s - mi}.$$

Summing over all $m \geq 1$, the total expected number of pruned subsets at step $s$, with successive overcount corrections added and subtracted depending on the overlap level, is:

$$\sum_{i=1}^{s-1} \sum_{m=1}^{\lfloor p\binom{n}{i}\rfloor} (-1)^m \binom{p\binom{n}{i}}{m}\binom{n - mi}{s - mi}. \tag{9}$$

To obtain the total expected number of subset evaluations over all steps, we sum the unpruned subsets across all sizes. At step $s = 1$, all $n$ singleton subsets are evaluated.

For $s \geq 2$, each subset of size $s$ survives pruning with probability $1 - p$, and only if it has not been eliminated due to overlap with smaller selected subsets. Thus, the expected number of evaluations at step $s$ is:

$$(1 - p) \cdot \left( \binom{n}{s} + \sum_{i=1}^{s-1} \sum_{m=1}^{\lfloor p \binom{n}{i} \rfloor} (-1)^m \binom{p \binom{n}{i}}{m} \binom{n - mi}{s - mi} \right).$$

The inner summation captures the pruning corrections from the inclusion–exclusion expansion in Equation (9). To ensure non-negativity of the expected number at each step, we take a maximum with zero.

Adding this to the evaluations from $s = 1$, the total expected number of subset evaluations is:

$$n + (1 - p) \sum_{s=2}^{n} \max \left( 0, \binom{n}{s} + \sum_{i=1}^{s-1} \sum_{m=1}^{\lfloor p \binom{n}{i} \rfloor} (-1)^m \binom{p \binom{n}{i}}{m} \binom{n - mi}{s - mi} \right).$$

$\square$

**Theorem 1** (Expected Time Complexity of Minimality-based Subset Search). *Consider a minimality-based subset search over $n$ nodes, where each subset is independently selected as a causal node set with probability $p$. Then, the expected number of subset evaluations is upper-bounded as in Lemma 1. Based on this bound, the expected time complexity grows approximately as*

$$O \left( n^{\lfloor \log_2 \left( \frac{1}{p} + 2 \right) \rfloor} \right).$$

*Proof.* From Lemma 1, the expected number of subset evaluations is:

$$n + (1 - p) \sum_{s=2}^{n} \max \left( 0, \binom{n}{s} + \sum_{i=1}^{s-1} \sum_{m=1}^{\lfloor p \binom{n}{i} \rfloor} (-1)^m \binom{p \binom{n}{i}}{m} \binom{n - mi}{s - mi} \right). \tag{10}$$

We approximate the pruning term by keeping only the $m = 1$ intersection term in the inclusion-exclusion sum, discarding the rest ($m \geq 2$), which largely cancel out due to alternating signs.

$$\binom{n}{s} - \sum_{i=1}^{s-1} p \binom{n}{i} \binom{n - i}{s - i}.$$

Using the fact that $\binom{n}{i} \binom{n-i}{s-i} = \binom{n}{s} \binom{s}{i}$ and $\sum_{i=1}^{s-1} \binom{s}{i} = 2^s - 2$, we simplify the above as:

$$\binom{n}{s} \left( 1 - p(2^s - 2) \right).$$

Substituting this into Equation (10), we obtain the following:

$$n + (1 - p) \sum_{s=2}^{n} \max \left( 0, \binom{n}{s} (1 - p(2^s - 2)) \right). \tag{11}$$

A subset of size $s$ contributes to the sum ($\sum_{s=2}^{n}$) only if the term with pruning factor is positive, i.e., the term $(1 - p(2^s - 2))$ applied to the positive binomial term $\binom{n}{s}$ is greater than zero:

$$1 - p(2^s - 2) > 0 \quad \leftrightarrow \quad s < \log_2 \left( \frac{1}{p} + 2 \right).$$

Here, let $s^* := \left\lfloor \log_2 \left( \frac{1}{p} + 2 \right) \right\rfloor$ be the largest such $s$ that satisfies this condition. Then, Equation (11) reduces to:

$$n + (1-p) \sum_{s=2}^{s^*} \binom{n}{s} (1 - p(2^s - 2)).$$

Since each binomial coefficient satisfies $\binom{n}{s} = O(n^s)$ and the term $(1 - p(2^s - 2))$ acts as a constant for each fixed $s$, the summation is upper-bounded as

$$\sum_{s=2}^{s^*} \binom{n}{s} (1 - p(2^s - 2)) = O(n^2 + \cdots + n^{s^*}) = O(n^{s^*}).$$

Thus, the total number of evaluations grows approximately as

$$n + (1-p) \cdot O(n^{s^*}) = O(n^{s^*}).$$

$\square$

**Remark 5.** *(This remark extends Remark 4.) Since the underlying search space contains $2^n$ possible subsets, the time complexity approaches $O(2^n)$ as $p$ becomes very small. This occurs when the term with pruning factor $1 - p(2^s - 2)$ remains positive up to $s = n$, which requires $p < \frac{1}{2^n - 2}$. However, such cases occur only infrequently in practice, especially considering that $n$ corresponds to the number of path nodes in a single transformer block, typically $2H + 2$.*

### F.2 PROOF OF THEOREM 2

**Theorem 2** (Causal Union Reference Reliability). *Consider a minimality-based subset search over $n$ nodes, where each subset is independently selected as a causal node set with probability $p$. Suppose that a collection of such sets, $V_{out}^{(j+1)} = \{V_i^{(j+1)}\}_{i=1}^k$, is identified as causal in the $(j+1)$-th block. Their union, denoted as $P = \bigcup_{i=1}^k V_i^{(j+1)}$, serves as the causal subpath reference for the minimality-based subset search in the $j$-th block. Let $s_{avg}$ denote the average size of the $k$ causal node sets in $V_{out}^{(j+1)}$. Then, the reliability of the resulting causal node set obtained using $P$ is given by:*

$$p + (1-p)\left(1 - \left(1 - \frac{s_{avg}}{n}\right)^k\right)^n \to 1 \tag{12}$$

*Proof.* To evaluate the reliability of using the union $\bigcup_{i=1}^k V_i^{(j+1)}$ as the causal subpath reference $P$ during the search in the $j$-th block, we consider the following two cases under which this strategy ensures a reliable outcome.

First, $P$ itself could have been selected as a causal node set during the search in the $(j+1)$-th block, independently with probability $p$, even though it was not explicitly selected because its subsets had already been included.

Second, if $P$ was not selected in this way, it must equal the full set of path nodes in the $j$-th block. In this case, $P$ must at least correspond to the case in Property 1, exemplified by Example 2.

The total probability of success across these two cases is given by:

$$p + (1-p) \cdot \hat{p}, \tag{13}$$

where $\hat{p}$ denotes the probability of the second case.

To compute $\hat{p}$, note that the probability that a single causal node set in $V_{out}^{(j+1)}$, with average size $s_{avg}$, contains a given node among the $n$ path nodes in the $j$-th block is:

$$\frac{s_{avg}}{n}.$$

Accordingly, the probability that at least one of the $k$ causal node sets contains a given node is:

$$1 - \left(1 - \frac{s_{avg}}{n}\right)^k.$$

Extending this, the probability that the $k$ causal node sets collectively cover all $n$ path nodes, i.e., that $P$ equals the full set of path nodes in the $j$-th block, is:

$$\hat{p} = \left(1 - \left(1 - \frac{s_{\text{avg}}}{n}\right)^k\right)^n.$$

Substituting this into Equation equation 13, the reliability can be expressed as follows:

$$p + (1 - p) \cdot \left(1 - \left(1 - \frac{s_{\text{avg}}}{n}\right)^k\right)^n.$$

To analyze convergence, define $x = \frac{s_{\text{avg}}}{n}$. Using the inequality:

$$(1 - x) \leq e^{-x},$$

we obtain:

$$\left(1 - (1 - x)^k\right)^n \geq \left(1 - e^{-xk}\right)^n.$$

Substituting back $x = \frac{s_{\text{avg}}}{n}$, the lower bound for $\hat{p}$ becomes:

$$\hat{p} \geq \left(1 - e^{-k \cdot \frac{s_{\text{avg}}}{n}}\right)^n.$$

Here, $k$ is equal to the number of subset evaluations selected with probability $p$ from the minimality-based subset search. As $n$ grows large, $k$ asymptotically follows the evaluation bound shown in Lemma 1, which converges to $n^{s^*}$. Therefore, $k$ can be approximated as:

$$k \approx p \cdot n^{s^*} = p \cdot n^{\left\lfloor \log_2\left(\frac{1}{p} + 2\right)\right\rfloor}.$$

Thus, the reliability is lower bounded as:

$$p + (1 - p) \cdot \left(1 - \left(1 - \frac{s_{\text{avg}}}{n}\right)^k\right)^n \geq p + (1 - p) \cdot \left(1 - e^{-k \cdot \frac{s_{\text{avg}}}{n}}\right)^n$$

$$\approx p + (1 - p) \cdot \left(1 - e^{-p \cdot n^{s^*} \cdot \frac{s_{\text{avg}}}{n}}\right)^n$$

We now analyze the asymptotic behavior of this lower bound as $n$ grows large, focusing on how the reliability converges under varying values of $p$.

Specifically, as $p \to 1$, i.e., when most combinations of path nodes are selected as causal node sets with probability close to 1, the minimality-based subset search algorithm tends to converge at step 1. That is, both $s^*$ and $s_{\text{avg}}$ converge to 1, and the expression simplifies as follows:

$$p + (1 - p) \cdot \left(1 - e^{-p}\right)^n \to 1$$

This is because, as $p \to 1$, the factor $(1 - p)$ approaches zero, while $(1 - e^{-p})^n$ vanishes as $n$ increases. As a result, only the $p$ term remains, and the expression converges to 1.

Conversely, as $p \to 0$, i.e., when the probability of selecting any combination of path nodes as a causal node set becomes vanishingly zero, the minimality-based subset search algorithm tends to converge at step $n$; thus, $s^* \to n$. In this case, $s_{\text{avg}}$ converges to $\frac{n}{2}$ for the following reason:

$$s_{\text{avg}} = \frac{\sum_{i=1}^{s^*} i \cdot \binom{n}{i}}{\sum_{i=1}^{s^*} \binom{n}{i}} \approx \frac{n \cdot 2^{n-1}}{2^n} = \frac{n}{2}$$

Therefore, the lower bound of the reliability simplifies as follows:

$$p + (1 - p) \cdot \left(1 - e^{-p \cdot n^n \cdot \frac{1}{2}}\right)^n \to 1$$

This is because the term $\left(1 - e^{-p \cdot n^n \cdot \frac{1}{2}}\right)^n$ converges to 1 as $n$ grows large, while $p$ goes to zero. Hence, the entire expression converges to 1.

Note that in our context, $n$ refers to the number of path nodes extracted from a single block. In typical transformer architectures, even with a single attention head, the number of path nodes per block is at

least 4 (i.e., $2H + 2$). Empirically substituting such moderate values of $n$ into the expression confirms that the lower bound still converges to 1, even when $n$ is not particularly large.

Furthermore, since the reliability lower bound is monotonic in $p$, it converges to 1 regardless of the specific value of $p \in (0, 1)$. In addition, because reliability represents a probability bounded between 0 and 1, the entire expression is also upper bounded by 1. Accordingly, the reliability satisfies the following tight bound:

$$p + (1 - p) \cdot \left( 1 - \left( 1 - \frac{s_{\text{avg}}}{n} \right)^k \right)^n \to 1$$

In conclusion, using the union of causal node sets from the previous block as the causal subpath reference in the current block yields a reliability that consistently converges to 1.  □

## G   IMPLEMENTATION DETAILS AND ADDITIONAL RESULTS

### G.1   COMMON EXPERIMENTAL SETUP

The code is attached as part of the supplementary material and is also accessible through an anonymized link[1]. All experiments were conducted using an RTX Quadro 6000 24GB GPU with PyTorch 2.1.0 and CUDA 11.8. For three language models (GPT2-xs[2], Pythia-14m[3], and Pythia-1b[4]) and two vision models (ViT-tiny and DeiT-tiny[5]), the internal decision paths were traced while keeping all pretrained parameters frozen. Note that for the OFFICEHOME dataset, only the classifier was fine-tuned to match the number of classes. In the case of Pythia, since it adopts a parallel residual structure where the block input is directly fed into both the attention and MLP components, unfolding is adapted accordingly to reflect this architecture; as a result, each block produces $H + 2$ paths instead of $2H + 2$. Since a decision path is considered meaningful only if the model produces the correct output for a given input, we sampled subset of the evaluation set that were correctly predicted; but the same evaluation set was used across all methods to ensure a fair comparison.

When using our proposed method with TOKEN RESAMPLING, to prevent incorrect interventions caused by coincidentally selecting token embeddings similar to the original input, we sampled 100 different resampling batches and computed the causal evaluation as the average output across these interventions. For $D_\alpha$ and $D_\beta$ in each block, we computed their values using the original block input when implementing the unfolded path nodes. Following the identification of causal paths, we utilized their union for further analysis, as justified by Theorem 2.

### G.2   HOW SELF-REPAIR IS COMPUTED

Building on the notion of self-repair introduced by the prior work (Rushing & Nanda, 2024), we define the self-repair score received by each attention head in a given layer as:

$$\text{self-repair} = \Delta logit - \Delta DE_{head}. \tag{14}$$

Here, $\Delta logit$ measures the change in the logit corresponding to the original decision after ablating the attention head. $\Delta DE_{head}$ measures the direct effect of the head, i.e., its original contribution to the decision logit without self-repair.

Specifically, the final residual stream, i.e., the classifier input, contains the sum of outputs from all heads across all layers. Accordingly, the contribution of each head can be directly approximated by projecting its output onto the classifier weight vector; this yields the direct effect of that head, $\Delta DE_{head}$. Furthermore, following the assumption in the prior work (Rushing & Nanda, 2024) that an ablated head contributes zero to the decision, we compute $\Delta DE_{head} = -DE_{head}$.

---

[1] https://tinyurl.com/neurips15566
[2] https://huggingface.co/AlgorithmicResearchGroup/gpt2-xs
[3] https://huggingface.co/EleutherAI/pythia-14m
[4] https://huggingface.co/EleutherAI/pythia-1b
[5] https://github.com/huggingface/pytorch-image-models

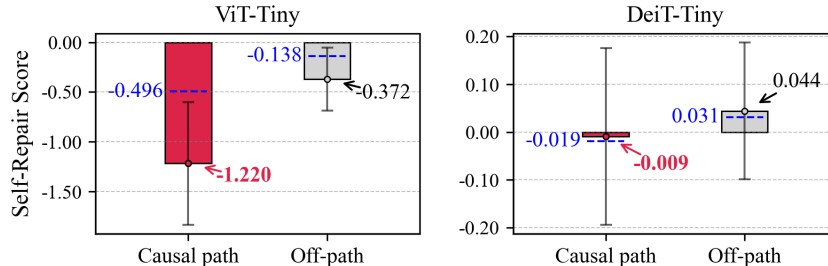

Figure A: **Self-repair scores on causal path vs. off-path components in vision models.** Each bar shows the mean (dot with arrow) and standard deviation (error bar); medians are shown as blue dashed lines. Lower scores indicate less self-repair. Results are averaged over IMAGENET and OFFICEHOME.

### G.3 WHAT SELF-REPAIR MEANS IN OUR CONTEXT

Self-repair, also referred to as backup behavior or a backup mechanism, serves a functional role in maintaining the model's behavior. In this section, we aim to clarify both its meaning and its implications in relation to causal paths.

In our context, the self-repair score reflects how much a target node receives backup from other nodes, thereby indicating its role as a backup receiver. In other words, there are two roles in a backup mechanism during ablation: a giver node, which supplies the compensatory signal, and a receiver node, which is affected by it. The self-repair score we measure captures the extent to which a node functions as a receiver in this process.

Specifically, for a node to act as a giver, it must be capable of supporting the decision itself; its contribution to the decision must be high.Given that the total logit for the decision is a fixed quantity, greater contribution by givers implies lesser contribution from receivers. In this sense, giver nodes involved in backup mechanisms may be seen as indicative of the causal mechanism itself. Therefore, if the discovered paths contain many receiver nodes, their self-repair scores will be high, indicating a lower contribution to the decision compared to other paths composed primarily of giver nodes; that is, such discovered paths are likely less causal. In contrast, if receiver nodes mostly lie outside the discovered paths, their self-repair scores within the paths will be lower than those outside, suggesting that the discovered paths contribute more directly to the decision than nodes outside the paths. Taken together, a low self-repair score indicates that the discovered path contributes more directly to the decision than components outside it, thereby serving as evidence of its causal nature.

### G.4 SELF-REPAIR ANALYSIS IN VISION MODELS

**Setup** As described in Section G.2, we measure the self-repair score of each attention head across layers. A head is considered to be on the causal path if it is included in any causal path at least once; otherwise, it is classified as off-path.

**Results** Extending the findings from the main script on language models, we examine whether the same pattern holds in vision models. As shown in Figure A, although self-repair behavior is inherently noisy, as previously reported, we still observe consistently lower self-repair scores on components included in the causal path. This suggests that, even in vision models, the identified causal paths correspond to components that play an essential and non-redundant role in the decision-making process of the model.

### G.5 CLASS-SPECIFIC PATH DISCOVERY

**Setup** We conducted experiments by removing the largest activated causal subpath (Top-1 in Figures B and 4) or the top three largest ones (Top-3 in Figures B and 4) for a given super-class. For IMAGENET, we selected a subset of 1,000 ImageNet classes and grouped them into four super-classes based on WordNet semantic similarity. The defined superclasses are as follows: **dog**, which includes Maltese, beagle, Saluki, Siberian husky, and golden retriever; **bird (oscines)**, consisting of brambling, robin, jay, and chickadee; **garment**, comprising kimono, cardigan, T-shirt, and sweatshirt; and **vehicle**,

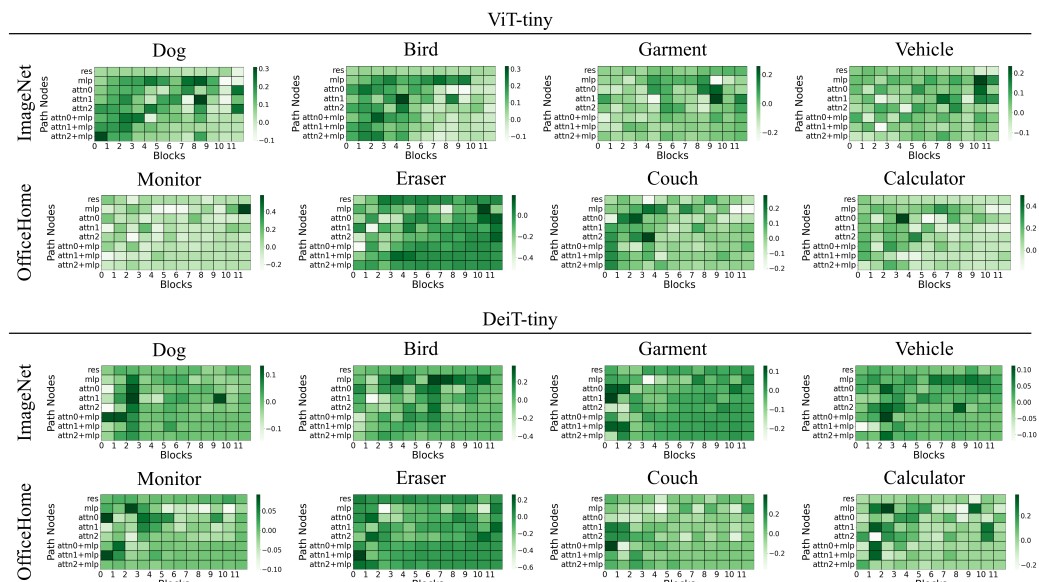

Figure B: **Class-specific causal path patterns.** The class-wise difference in average causal path ratios between a target class and all classes, highlighting class-specific paths. Here, res, mlp, and attn# indicate residual, MLP, and attention paths from head #, respectively.

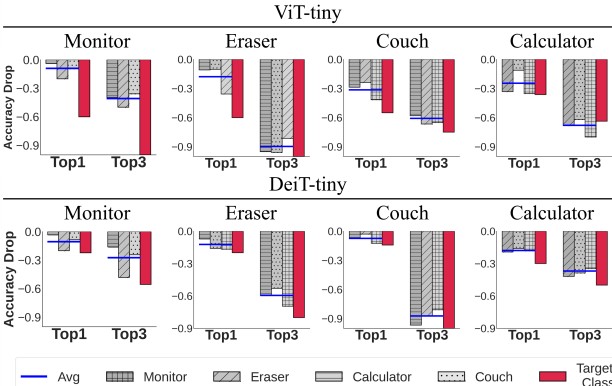

Figure C: **Causal paths uniquely activated for specific classes on OFFICEHOME.** Accuracy drop when ablating the most class-specific path, showing selective reliance by each class.

which includes cab, minibus, moving van, police van, and school bus. In contrast, in OFFICEHOME, the visual dissimilarity between classes is too large to form meaningful super-classes, so each class was treated as its own super-class.

**Results** The activation ratio patterns of the causal paths for each super-class can be seen in Figure B. The stronger the color contrast, the more exclusively the corresponding causal subpath is activated for that specific super-class. Furthermore, similar to the main script results, Figure C shows that in OFFICEHOME, ablating the most activated causal subpath for each super-class leads to a significantly larger accuracy drop for that super-class compared to the average drop across others. This indicates the existence of causal subpaths that are specifically responsible for class-specific decisions. However, when more causal subpaths are ablated, the performance drop becomes more uniform, suggesting that, since each class is treated as its own super-class, the model does not rely heavily on a large number of unique subpaths per class. In other words, critical components for fine-grained decisions appear to be more sparsely encoded within the model.

| | Model | KNOWNS1000 | | | T-REX | | |
|---|---|---|---|---|---|---|---|
| | | Hit. (↑) | Faith. (↑) | Spars. (↓) | Hit. (↑) | Faith. (↑) | Spars. (↓) |
| $NT_1$ | GPT2-xs | 0.0000 | 0.0004 | 0.8244 | 0.0000 | 0.0005 | 0.8261 |
| | Pythia-1b | 0.0000 | 0.0001 | 0.2197 | 0.0000 | 0.0001 | 0.2197 |
| | Pythia-14m | 0.0000 | 0.0009 | 0.8964 | 0.0000 | 0.0012 | 0.8964 |
| $NT_{10\%}$ | GPT2-xs | 0.0000 | 0.0001 | 0.6960 | 0.0000 | 0.0001 | 0.6870 |
| | Pythia-1b | 0.0000 | 0.0001 | 0.1503 | 0.0000 | 0.0001 | 0.1503 |
| | Pythia-14m | 0.0000 | 0.0002 | 0.8526 | 0.0000 | 0.0010 | 0.8526 |
| $ET_{all}$ | GPT2-xs | 0.1111 | 0.1442 | 0.9868 | 0.6666 | 1.0643 | 0.9868 |
| | Pythia-1b | 0.4699 | 0.2016 | 0.9998 | 0.0000 | 0.0002 | 0.9629 |
| | Pythia-14m | 0.0000 | 0.0007 | 0.9737 | 0.0000 | 0.0016 | 0.9737 |
| $ET_{cls}$ | GPT2-xs | 0.5556 | 0.3922 | 0.9868 | 0.6078 | 0.9543 | 0.9853 |
| | Pythia-1b | 0.5904 | 0.4988 | 0.9964 | 0.4686 | 0.3459 | 0.9903 |
| | Pythia-14m | 0.1429 | 0.1639 | 0.9901 | 0.5195 | 0.4855 | 0.9967 |
| ET-$IG_{all}$ | GPT2-xs | 0.5000 | 0.5239 | 0.9840 | 0.5922 | 0.9281 | 0.9816 |
| | Pythia-1b | 0.9759 | 0.9815 | 0.9995 | 0.9617 | 0.9369 | 0.9996 |
| | Pythia-14m | 0.3571 | 0.2473 | 0.9988 | 0.6190 | 0.8802 | 0.9992 |
| ET-$IG_{cls}$ | GPT2-xs | 0.6111 | 0.4329 | 0.9794 | 0.5534 | 0.8388 | 0.9782 |
| | Pythia-1b | 0.9759 | 0.9089 | 0.9993 | 0.9669 | 0.9528 | 0.9995 |
| | Pythia-14m | 0.5893 | 0.4038 | 0.9984 | 0.7143 | 0.9569 | 0.9991 |
| **CPT** (ours) | GPT2-xs | **1.0000** | 0.3742 | 0.9004 | **0.9903** | 0.7260 | 0.8929 |
| | Pythia-1b | 0.9518 | 0.2593 | **0.7246** | 0.9843 | 0.6192 | **0.6966** |
| | Pythia-14m | 0.9821 | **0.3857** | 0.9848 | 0.9870 | **0.9218** | 0.9851 |

Table A: **Quantitative results (language).** Detailed results three models on two datasets. The average performance of each method across models and datasets in this table is averaged in Table 2. Hit. and Spars. range between 0 and 1, while Faith. takes non-negative values (i.e., 0 or greater). A Faith. score greater than 1 indicates that the logit for the original decision was amplified by that factor.

| | Model | IMAGENET | | | OFFICEHOME | | |
|---|---|---|---|---|---|---|---|
| | | Hit. (↑) | Faith. (↑) | Spars. (↓) | Hit. (↑) | Faith. (↑) | Spars. (↓) |
| $NT_1$ | ViT-tiny | 0.0014 | 0.0016 | 0.7618 | 0.0140 | 0.0252 | 0.7943 |
| | DeiT-tiny | 0.0000 | 0.0019 | 0.6608 | 0.0265 | 0.0258 | 0.6933 |
| $NT_{10\%}$ | ViT-tiny | 0.0000 | 0.0036 | 0.0873 | 0.0202 | 0.0281 | 0.0721 |
| | DeiT-tiny | 0.0000 | 0.0009 | 0.0844 | 0.0109 | 0.0206 | 0.0759 |
| $ET_{all}$ | ViT-tiny | 0.2816 | 0.1693 | 0.9997 | 0.4022 | 0.3776 | 1.0000 |
| | DeiT-tiny | 0.5749 | 0.2833 | 1.0000 | 0.5229 | 0.4361 | 1.0000 |
| $ET_{cls}$ | ViT-tiny | 0.0695 | 0.0435 | 0.9573 | 0.2147 | 0.1889 | 0.9291 |
| | DeiT-tiny | 0.4561 | 0.2541 | 0.9928 | 0.3105 | 0.2464 | 0.9810 |
| ET-$IG_{all}$ | ViT-tiny | 0.1166 | 0.1490 | 0.9816 | 0.0100 | 0.0007 | 0.9997 |
| | DeiT-tiny | 0.6867 | 0.3444 | 0.9457 | 0.5500 | 0.4306 | 0.9997 |
| ET-$IG_{cls}$ | ViT-tiny | 0.1333 | 0.1713 | 0.9378 | 0.0263 | 0.0123 | 0.9207 |
| | DeiT-tiny | 0.6578 | 0.3039 | 0.9866 | 0.3833 | 0.3123 | 0.9545 |
| **CPT** (ours) | ViT-tiny | **0.9743** | **0.2099** | 0.8114 | 0.9506 | **0.4835** | 0.7218 |
| | DeiT-tiny | 0.9675 | 0.1175 | **0.6812** | 0.9630 | 0.3855 | **0.6975** |

Table B: **Quantitative results (vision).** Detailed results two models on two datasets. The average performance of each method across models and datasets in this table is averaged in Table 3. Hit. and Spars. range between 0 and 1, while Faith. takes non-negative values (i.e., 0 or greater). A Faith. score greater than 1 indicates that the logit for the original decision was amplified by that factor.

## G.6 Detailed Quantitative Comparisons

**Setup** To adapt the baselines to a decision-focused path tracing framework, we explored the model internals in a backward chaining manner and assigned each discovered decision path to one of the following types: residual-only, attention-only, MLP-only, or attention+MLP-only.

Let $y$ with decision $c^*$ denote the original model output, and let $y'$ with decision $c'^*$ be the output when all components not on the selected path (as determined by each method) are pruned. For **Hit.** (hit rate) score, we measured the proportion of the dataset where $c^* = c'^*$. For **Faith.** (faithfulness) score, we measured the average ratio $y'^{(c^*)}/y^{(c^*)}$ across the dataset. For **Spars.** (sparsity) score, we computed the average ratio of FLOPs used by the components along the selected decision path to the total FLOPs of the original model. Note that even if an attention-only path is not selected, if an attention+MLP path is chosen, the attention layer's FLOPs are included in the calculation.

**Results** The results across models and datasets—those averaged in Tables 2 and 3 of the main script—are presented in Tables A and B. As previously mentioned, our method uniquely achieves a near-perfect hit rate, reliably identifying the causal path responsible for the original decision. While some individual cases show relatively higher faithfulness scores under other methods, the consistently low hit rates point to a critical limitation: even if a method increases the logit value for the original decision index, the fact that the final prediction changes implies that it has increased the logits for other indices even more. This means that the identified path is not truly associated with the intended decision, but rather explains a different one—leading to a misattribution. In fact, when examining the sparsity scores, these methods often utilize nearly the entire internal structure of the original model, suggesting minimal pruning and limited explanatory focus.

For instance, $ET_{all}$ applied to GPT2-xs on T-REX marginally increased the logit of the original decision index (1.0643). However, the hit rate remained at 0.6666, indicating that the method amplified other logits even more and ultimately changed the decision.

Similarly, for DeiT-tiny on IMAGENET, both $ET_{all}$ and $ET_{cls}$ may appear to better recover the original logit value compared to ours. However, while our method explicitly selects a decision path that preserves the original decision (achieving a hit rate close to 1), their selected paths maintain the original decision in only about half the cases. Notably, these results occur despite their extremely high sparsity scores, indicating that almost the entire model structure was utilized.

Additionally, for Pythia-1b on KNOWN100 and T-REX, $ET$-$IG_{all}$ and $ET$-$IG_{cls}$ achieve performance comparable to ours only in this specific case. However, their sparsity scores (close to 1) indicate that such results do not stem from selecting the truly decision-critical components, but rather from consistently attributing explanations to nearly the entire model. As a consequence, these outcomes provide little insight into where the relevant information for each decision actually resides within the model.

## G.7 Actual Wall-Clock Time

We aim to explain why exploring all possible path combinations was intractable in prior work, and why achieving tractability (on average) through our method is such a crucial advancement, as evidenced by the actual wall-clock time in Table C.

For each model, every block contains $n$ paths—typically $2H + 2$, as shown in Equation (2)—and the number of possible combinations is $2^n$ (including the empty set for simplicity). Across $D$ blocks, the total number of possible paths from input to decision amounts to $2^{n \times D}$. This represents the total search space of a greedy search strategy required to guarantee reliable path tracing for a decision. The scale of this space is enormous and practically infeasible to exhaustively explore.

Consequently, NT restricted its exploration to combinations only at the level of attention and MLP outputs, while ET considered only pairwise combinations (ignoring combinations of three or more). Here, although ET-IG requires slightly more time than ET, its runtime is similar and is therefore omitted. Still, these reduced spaces remain tiny compared to the total search space, which inevitably results in either large explanatory errors or explanations that indiscriminately cover the whole model instead of concentrating on the critical components.

| Model | Total Search Space $(2^{n \times D})$ | NT $(\frac{3 \times D}{2^{n \times D}} \ll 0.0001\%)$ | ET $(\frac{\binom{(H+2) \times D}{2}}{2^{n \times D}} \ll 0.0001\%)$ | CPT $(\approx 100\%)$ |
|---|---|---|---|---|
| GPT2-xs | $2^{14 \times 6}$ | 0.16sec $\pm$ 0.03 | 1.80sec $\pm$ 0.02 | 17632.17sec $\pm$ 2113.36 |
| Pythia-1b | $2^{10 \times 16}$ | 1.31sec $\pm$ 0.17 | 8.49sec $\pm$ 0.36 | 3114.00sec $\pm$ 930.95 |
| Pythia-14m | $2^{6 \times 6}$ | 0.13sec $\pm$ 0.01 | 1.76sec $\pm$ 0.01 | 10.04sec $\pm$ 0.86 |
| ViT-tiny | $2^{8 \times 12}$ | 0.96sec $\pm$ 0.04 | 28.07sec $\pm$ 3.05 | 734.04sec $\pm$ 283.12 |
| DeiT-tiny | $2^{8 \times 12}$ | 1.00sec $\pm$ 0.04 | 30.14sec $\pm$ 4.09 | 564.00sec $\pm$ 86.76 |

Table C: **Actual wall-clock time of search.** The "Total Search Space" column represents all possible combinations of internal path nodes across layers, where $n$ is the number of paths per block, $D$ is the number of blocks, and $H$ is the number of heads per block. Here, $n$ is generally $2H + 2$ because residual connections are applied sequentially to both attention and MLP. However, in the Pythia family, residual connections are applied in parallel, so $n$ becomes $H + 2$. The percentages in parentheses indicate each method's coverage over the total search space. For NT, only the combinations of the attention output and MLP output within each block are considered independently, resulting in $3 \times D$ search trials. For ET, only pairwise combinations among attention heads, MLP, and residual connections—$(H + 2) \times D$ in total—are considered, resulting in $\binom{(H+2) \times D}{2}$ search trials.

In contrast, our method is the first to cover the entire total search space while enabling polynomial-time exploration on average. Furthermore, compared with other methods in terms of search-space coverage, the actual wall-clock time of our method is reasonably small. Nevertheless, as we note in the Section 4, the reduced time scale inevitably grows as $n$ increases, and thus further improvements will be necessary to scale our method to even larger models in future work.

### G.8 COMPARISON WITH IOI GROUND TRUTH

We highlight the connection between our causal path tracing and the ground-truth circuits manually annotated in prior work (Wang et al., 2022) for the IOI (Indirect Object Identification) task. Since the IOI task also requires identifying the critical circuits responsible for a model's decision, it is closely related to our approach, and our method achieves high precision on this task:
CPT: 0.727, $NT_1$: 0.146, $NT_{10\%}$: 0.392, $ET_{all}$: 0.157, ET-$IG_{all}$: 0.154.

Specifically, the ground-truth circuits for IOI were generated using GPT2-small (Radford et al., 2019). Because the total search space in GPT2-small is extremely large ($2^{26 \times 12}$), we evaluated performance on 20 randomly selected samples. Furthermore, since the IOI task does not involve class labels, we exclude $ET_{cls}$ and ET-$IG_{cls}$ from this comparison. We also employed a weakened variant of CPT: as this model has a relatively large number of heads ($H = 12, n = 26$), our minimality-based path tracing can get stuck in worst-case traps, resulting in prohibitively long runtimes. To address this, we restricted the search to subsets of size up to three. Even under this constraint, our method shows strong alignment with the IOI ground-truth circuits, as evidenced by its high precision.

Notably, while other methods achieved recall close to 1, they tended to return nearly the entire model as a decision path rather than pinpointing the critical components (as indicated by the high sparsity scores in Tables 2 and 3). For this reason, we regard precision as the more meaningful metric for evaluating the connection to the IOI task.

## H APPLICATIONS AND FUTURE EXTENSIONS TO SEQUENTIAL GENERATION

While our experiments primarily focus on interpretability-oriented phenomena (e.g., identifying causal paths), the proposed algorithm can be naturally extended to practical tasks such as model debugging and pruning. For example, by tracing the causal path of a misclassified sample, one can inspect where the model relies on non-causal components to make incorrect decisions. Moreover, pruning nodes that are not included in the causal paths of given samples—either during inference or fine-tuning—has the potential to reduce model size and computational cost without harming decision quality.

Also, in line with prior research (e.g., Meng et al. (2022a); Syed et al. (2024)), our algorithm primarily focuses on classification and single-token generation tasks. A key challenge in extending causal

path analysis to complex generative tasks, such as multi-token generation, is the potential existence of multiple causal paths, along with the need to understand their interactions. Accordingly, we leave the question of how to identify and unify causal paths for multi-token generation as an open direction for future work. Addressing this challenge could lay the groundwork for training models that better adhere to causal reasoning, while also supporting more systematic auditing and refinement of non-causal decision mechanisms.

