# OpenReview forum: "Causal Path Tracing in Transformers"
_ICLR.cc/2026/Conference — Submitted to ICLR 2026_

### Official Review · Reviewer_BTiY · 2025-10-27

**Soundness:** 2
**Presentation:** 3
**Contribution:** 2
**Rating:** 4
**Confidence:** 3

**Summary:**

The paper introduces a causal path tracing framework for explaining decisions of transformer models. The path-level approach allows for causal referencing, a technique that avoids incorrect explanations due to the self-repair behavior of these models. To make the path-level approach feasible, it employs a minimality-based subset search for identifying causal paths. Experimentally, the paper finds that self-repair happens less frequently along the found causal paths and that these paths accurately recover the model decision.

**Strengths:**

The paper is generally well-written. It extends a recent line-of-work on causal interpretability of transformer models, certainly a fundamental topic in the current AI literature. It has a clear story of making path tracing methods efficient by doing a minimal subset search which is effective for keeping the run-time manageable in practice. Moreover, it provides evidence on small-ish models to corroborate main claims (e.g. that self-repair happens outside of "causal paths").

**Weaknesses:**

The main weakness of the paper is that it lacks clarity and mathematical rigor.

Many of the main notions in the paper are defined rather loosely and their relation to standard graphical or causal notions seems unclear to me. For example, even the main definition of a path as "... a sequence of node sets connected through [...] edges" is not fully clear to me. Similarly, causal notions are also introduced quite loosely and hand-wavy. Connections to Pearl's theory of causality (causal paths) are not formalized (despite Pearl being cited as source of inspiration) and remain vague.

The treatment of the main algorithmic improvement (Algorithm 1) also uses theoretical notions too loosely. While the idea to work with minimal subsets and thus improve the practical run-time is nice (and it's intuitive that this yields run-time improvements), I find Theorem 1 to be too general to give much actual insight in the context of transformer models (with the artificial assumption that causal node sets are independently Bernoulli distributed). Moreover, it is claimed that the problem at hand is "NP-complete" without this being corroborated by a proof. The statement "... this subset search problem is NP-complete" is a bit of an inaccuracy in itself with NP containing decision problems and not search problems, but apart from that it is per se not clear (despite being of course plausible) whether the studied problem is NP-hard (for transformer models) and such a claim would necessitate a formal problem statement and a reduction from an NP-hard problem.

This also makes it hard to judge the merit of the method in general and compared to related work. I am not an expert in explainability (also reflected in my confidence score) and this may add to this issue, however, it remains at least partially unclear what (practical) advantages the proposed methods offer. Self-repair scores are low for "causal paths", but it would be interesting to learn more about the concrete insights this delivers or where this shows related interpretability methods to be lacking and giving (causally) incorrect interpretations.

**Questions:**

1. Regarding the definition of paths: Why are node sets connected and not individual nodes and what does "connected" mean here precisely? How does this notion compare to the standard definition of paths in directed graphs (as it seems to differ from that)?

2. Regarding causal notions: Is it possible to more precisely characterize the connection to Pearls causal theory? For example, can definition 4 be rephrased with regard to causal effects and the do-notation (e.g. 4a and 4b)?

3. Regarding the NP-hardness of causal subset search: Can you provide a proof sketch of the NP-hardness/NP-completeness of this problem (for transformer architectures)?


Suggestions (things I stumbled over, you may or may not heed these):

- Def. 1: "each node is deterministically computed from its parent nodes. [...] conditionally independent of its non-descendants": clarify how "conditionally independent" can be interpreted despite deterministic relations
- Def. 4: "all nodes between V and the output", this (and other notions) can be made more precise by building on standard graph terminology, maybe you mean all nodes on a path between a node in V and the output?
- Citations: you cite the arxiv version of some papers that have been published (eg Rushing, Nanda 2024 and Zhang, Nanda 2023)

---

> ### Author Response · Authors · 2025-11-21
>
> We sincerely thank the reviewer for the valuable and constructive feedback, which helped improve the clarity and rigor of our paper. Below, we provide our responses to each comment, along with the corresponding revisions made in the manuscript. Although some definition numbers have been updated in the revised manuscript, we retain the original numbering only in the response titles (e.g., [R1]~) to ensure easy mapping to the reviewer’s comments. Furthermore, we note that the main manuscript and the appendix (which were originally submitted as separate files) have now been merged into a single file.
>
> ---
>
> # [R1] Regarding the definition of paths
>
> ## [R1-1] Why we use node sets rather than individual nodes
>
> We consider node sets (rather than individual nodes) as causal candidates because, in general, the effect on a child node cannot always be attributed to individual parents independently; **interactions among parent nodes can jointly produce an effect** that no single node can generate alone.
>
> To illustrate this, consider the following simple example. Suppose the output representation is two-dimensional, and the model predicts the correct decision whenever the representation lies in the first quadrant. Assume that intervening on all parent nodes sets the output representation to the baseline vector $(0, 0)$, and that two parent nodes $v_1$ and $v_2$ contribute additively as follows:
>
> $v1$ adds $(-1, 2)$, $v2$ adds $(2, -1)$. The possible parent node sets are {$v_1$}, {$v_2$}, and {$v_1, v_2$}. If we evaluated a cause individual node by individual node, then:
>
> For {$v_1$}, $(0,0)+(-1, 2)=(-1, 2)$ is not in the first quadrant.
>
> For {$v_2$}, $(0,0)+(2, -1)=(2, -1)$ is not in the first quadrant.
>
> Hence, neither $v_1$ nor $v_2$ individually suffices to recover the original decision. However, since the internal computation within a transformer contains no unobservable latent confounders, it cannot be the case that the decision has no internal cause.
>
> When we evaluate node sets, we find:
>
> For {$v_1, v_2$}, $(0,0)+(-1, 2)+(2,-1)=(1, 1)$, which is in the first quadrant.
>
> Thus, the node set {$v_1, v_2$} is a valid cause of the decision—even though **neither node is a cause in isolation**.
>
> This example demonstrates that parent–parent interactions are essential and that treating each node independently would incorrectly conclude that “no cause exists.” For this reason, our causal search operates over node sets rather than individual nodes. We **added this clarification as Remark 2 in the revised manuscript.**
>
> ## [R1-2] For the meaning of "connected"
>
> By saying that two node sets are connected, **we intended to indicate that there exist direct computational dependencies, i.e., edges, between nodes in the first node set and nodes in the second node set.** We note that the term “connected” can be understood in two different ways—either as requiring every node in the first set to have an edge to every node in the second set, or as requiring only the existence of at least one such edge. **In our setting, however, this distinction does not arise due to the structure of the unfolded transformer graph.** As shown in the **“Path Node’’ region of Figure 1,** all path nodes within a block feed into a single block-output node, which then serves as the parent of all path nodes in the subsequent block. Consequently, **whenever two node sets are connected, every node in the first set influences every node in the second set through this structure.**
>
> ---
>
> # [R2] Regarding the rephrasing via do-notation
>
> We not that the **intervened node set $V'$ and $\hat{V}'$ can be naturally replaced with Pearl’s do-operator.** Accordingly, the two conditions in the definition of a causal node set can be written as follows:
>
> - Necessity: $\arg\max_i \mathbb{P}_i(y |  V, \hat{V}, P) \ne \arg\max_j \mathbb{P}_j(y |  do(V), do(\hat{V}), P)$,
> - Sufficiency: $\arg\max_i \mathbb{P}_i(y | V, \hat{V}, P) = \arg\max_j \mathbb{P}_j(y |  V, do(\hat{V}), P)$,
>
> where $\mathbb{P}_i(y|\cdot)$ equals $y^{(i)}$, the logit corresponding to index $i$. We added this to the main manuscript.
>
> ---
>
> # [R3] Regarding the NP-hardness of causal subset search
>
> We thank the reviewer for pointing this out, and we acknowledge that **our original phrasing was inaccurate. Therefore, we removed the inaccurate phrase “NP-complete.”** Our intended meaning was only that the naïve enumeration of all subsets requires an exponential search space. **This, however, does not contradict our claim that our algorithm achieves average-case polynomial behavior:** the minimality condition prunes the vast majority of supersets early, resulting in polynomial-time performance both in practice and under our theoretical expectation model. Our efficiency claim therefore concerns the average-case runtime of our algorithm, rather than the computational complexity of the underlying exhaustive subset search.

---

> ### Author Response · Authors · 2025-11-21
>
> # [R4] About Def 1 (Transformer as Causal Graph)
>
> We thank the reviewer for their careful reading. We agree that the term “deterministic” may appear to conflict with the statement of conditional independence. We have therefore revised the phrasing as follows: **“Each node depends only on its parent nodes. Once its parents are fixed, it is conditionally independent of its non-descendants.”**
>
> ---
>
> # [R5] About Def 4 (Causal Node Set)
>
> We thank the reviewer for this helpful suggestion. We agree that our original wording was imprecise and have revised the expression “all nodes between $V$ and the output" to **"all nodes except the ancestors of $V$".**
>
> ---
>
> # [R6] About Citations
>
> Thank you for the suggestion. We had previously cited these works based on their Google Scholar entries, which still list the arXiv versions. We updated the citations to reflect the officially published versions where available.

---

> > ### Comment · Reviewer_BTiY · 2025-11-25
> >
> > Thanks for the detailed response and the improvements to the manuscript!
> >
> > My general assessment of the submission remains unchanged and when reading the other reviews (e.g. from reviewer Kn3T) it seems that the sometimes vague definitions not only confused me. Therefore, I maintain my original score.
> >
> > Small nitpick:
> > line 261: "Although such exponential growth is unavoidable in the worst case, ..." I suggest to weaken that statement (to "likely unavoidable" or something similar)

---

> > > ### Author Response · Authors · 2025-11-25
> > >
> > > We sincerely thank the reviewer for the follow-up comment and for acknowledging the improvements made in the revision.
> > >
> > > We agree that the earlier draft had presentation issues that made several definitions appear vague, and **we revised those parts thoroughly** in response to all reviewers' comments.
> > >
> > > In particular, we reorganized the ordering of definitions and added an overview paragraph in Sec. 2, explicitly specified edge directions in Def. 2, introduced a do-notation–based formulation in Def. 4, clarified our design choices through Remark 2 and Remark 3, moved Def. 7 from the appendix to the main text, and corrected inaccurate wording (e.g., "possibly not minimal" in Property 1 and "NP-complete" in the main text), among other related refinements.
> > >
> > > From the overall discussion across the reviews, it seems that **the substantive aspects of our contribution—the methodology, theoretical analysis, and empirical findings—were not points of disagreement,** and that the concerns primarily arose from clarity and presentation.
> > >
> > > We have therefore **substantially improved the exposition, reorganized key definitions, and provided concrete examples and illustrations** to resolve the earlier ambiguity.
> > >
> > > We hope that the clearer presentation now makes the contributions more accessible and highlights the value of our approach for explainable and safe transformer-based systems.

---

### Official Review · Reviewer_a9Ri · 2025-10-28

**Soundness:** 4
**Presentation:** 3
**Contribution:** 2
**Rating:** 6
**Confidence:** 2

**Summary:**

The authors present a new technique for tracing the reasoning behind Transformer-based decisions by identifying their core causal pathways. The method operates by deconstructing Transformer blocks into a graphical representation of path nodes. It then employs a minimality-guided causal search across these blocks to isolate the precise chain of operations responsible for a specific output. Experimental validation confirms the method's efficiency and its superiority over existing baselines, proving that the identified pathways are integral to the model's decision-making process.

**Strengths:**

1. This work tackles a core issue in modern AI which is of significant current interest to the research community.
2. The paper's construction is logical, and the core ideas are articulated with clarity.
3. A key positive finding is the strong empirical evidence showing that the discovered causal paths are non-trivial and genuinely drive the model's predictions.

**Weaknesses:**

1. A significant limitation is the gap between the proposed interpretability method and its concrete application. The authors do not persuasively demonstrate how this tracing technique translates into tangible benefits for downstream tasks. The evaluation is more focused on the phenomenon of interpretability itself, rather than its practical consequences.
2. The methodological contribution regarding the average-case polynomial-time search appears derivative. It leans heavily on the causal minimality condition from the established Halpern–Pearl framework, which makes the proposed search algorithm feel more like an incremental extension than a novel breakthrough.

**Questions:**

1. How do the authors envision this framework bridging the gap to real-world applications? For example, can it be practically employed to diagnose a model's erroneous prediction in a specific case, or could it streamline the process of auditing a model for biased decision-making?

---

> ### Author Response · Authors · 2025-11-21
>
> We sincerely thank the reviewer for the valuable and constructive feedback, which helped improve the clarity and rigor of our paper. Below, we provide our responses to each comment, along with the corresponding revisions made in the manuscript. Although some definition numbers have been updated in the revised manuscript, we retain the original numbering only in the response titles (e.g., [R1]~) to ensure easy mapping to the reviewer’s comments. Furthermore, we note that the main manuscript and the appendix (which were originally submitted as separate files) have now been merged into a single file.
>
> ---
>
> # [R1] For Bridging Gap to Real-world Application
>
> We agree with the reviewer, and as discussed in Appendix Section H, **we view model debugging and pruning as the main application directions of our framework.** For future work, once the internal cause of an incorrect decision is identified, an important next step is to investigate how such faulty components can be corrected. **We believe this line of research can naturally extend to debugging methods that mitigate hallucinations** by modifying on the identified causal structures. In addition, if a model is intended to serve a specific task, one can prune the model by retaining only the causal paths that are relevant to that task. **We believe that this suggests a promising direction for building more cost-efficient, task-specialized models.**
>
> ---
>
> # [R2] For Methodological Contribution
>
> We would like to clarify that while our method builds on the Halpern–Pearl (HP) framework, our core contributions extend far beyond it. The HP framework provides a criterion for determining whether a variable is a cause, **but it does not address how to efficiently identify such causes in complex systems such as transformers.**
>
> Even when adopting the HP criterion, searching for causal node sets requires evaluating all subsets of parent nodes. Because this entails checking an exponential number of combinations, the resulting procedure does **not** admit polynomial-time complexity. **The HP framework itself does not provide any algorithmic strategy to avoid this non-polynomial behavior.**
>
> Moreover, when multiple causal node sets exist within a block, evaluating their upstream or downstream influence becomes even more computationally demanding, since each candidate set must be examined independently. This again leads to exponential time complexity, making end-to-end causal tracing infeasible.
>
> In contrast, our work introduces two essential components that make causal evaluation practical:
>
> - **Efficient within-block search.**
>
>     By leveraging the minimality condition, we design an algorithm that systematically prunes the exponential search space. This is not a heuristic; we formally prove that the algorithm achieves **average-case polynomial time**, which is a central technical contribution of our work.
>
> - **Scalable block-wise composition.**
>
>     We extend causal evaluation across blocks without incurring non-polynomial overhead. Importantly, we provide a theoretical guarantee that **the reliability of causal tracing is preserved under this composition**, enabling efficient end-to-end evaluation.
>
>
> These components are crucial for making causal tracing in transformers computationally tractable. Without them, adopting any causal criterion still leaves the computational problem intractable. As summarized in Table 1, **no prior work provides both theoretical tractability and empirical coverage of full transformer internals. Our method is the first to achieve both.**
>
> Additionally, by **unfolding each transformer block into meaningful causal paths,** our method provides interpretability value by elucidating how decisions are formed internally.

---

> > ### Comment · Reviewer_a9Ri · 2025-11-24
> >
> > I would like to thank the authors for their detailed response. I find that they have addressed my concerns, and I encourage them to incorporate this discussion into the revised manuscript. However, given my limited technical expertise in this specific area (as reflected in my confidence rating), I defer to the judgment of the other reviewers regarding the final decision.

---

> > > ### Author Response · Authors · 2025-11-25
> > >
> > > We sincerely thank the reviewer for the positive follow-up and for confirming that the earlier concerns have been addressed. We will incorporate the corresponding clarifications into the revised manuscript as suggested. We appreciate the reviewer's thoughtful consideration throughout the process.

---

### Official Review · Reviewer_Ruuc · 2025-10-30

**Soundness:** 2
**Presentation:** 1
**Contribution:** 2
**Rating:** 2
**Confidence:** 3

**Summary:**

This work focuses on interpreting the internal mechanism of a deep neural network architecture known as transformers via causal referencing, iteratively evaluating each component conditioned on priorly identified causal components along direct computational dependencies. Particularly, the paper addresses the issue of combinational complexity in path-level patching for causal referencing by proposing an efficient framework for tracing causal paths within each block of a transformer. Each block is treated as a path node and the whole transformer can be viewed as a causal graph. It introduces a minimality-based subset search strategy for identifying all possible causal path node combinations per block. This strategy is claimed to reduce the exponential complexity to polynomial time on average.

**Strengths:**

-	The paper attempts to solve an NP-complete problem.
-	It gives the first path-level patching method that feasibly enumerates all decision paths given an output.
-	It uses some simple tricks to rewrite the expression of the nonlinear function in a transformer to gain computational advantages.
-	The experiment supports that the search algorithm can still be useful in certain language models despite an unknown $p$ in Theorem 1.
-	The experiment shows some interesting use cases of the causal node sets.

**Weaknesses:**

- The writing needs significant improvement.
    - The paper does not define causal node sets before it uses this term in the definition of causal path.
     - The paper mentions things out of the blue without context. For example, line 67 says ‘self-repair occurs primarily…’, this is not clear about what the message is without describing what self-repair means. It then goes on to say ‘thus, the path contains information essential…’ to emphasize how important the causal path is. It is hard to understand the connection between the two statements. Subsequently, in line 069, it suddenly describes how causal paths are uniquely associated with specific classes. It is unclear what these specific classes mean. It implicitly suggests classification as the main goal of the transformer.
     - Definition 4 becomes confusing after reading lines 128-131. Please see the questions for details.
     -  Lines 135-136 give a very confusing statement regarding the definition of causal node set and the definition of causal node set (possibly not minimal). First, by definition 4, if $V$ is not minimal, $V$ is not a causal node set. The paper uses the same term with the addition ‘(possibly not minimal)’ to describe a set that does not meet the conditions of Definition 4.
     - Line 134 mentions ‘sufficient intervention’, but it is defined in line 147.
     - Definition 8 is not even mentioned in the main paper when it is used in line 301 as a part of Theorem 2.
- Some claims are not well-supported. Many definitions are introduced to support the design choice, but not justified why the definitions are necessary to achieve the goal.
- The last term named ‘Attention + MLP (H paths)’ in equation 2 is inconsistent with the derivation provided in Appendix E. The term $\frac{b_{oa}W_{ln2}^{\top} W_{lm}^{\top}  \mathbin{\circ} D_{B} W_{om}^{\top} }{H}$ is missing from the equation in the main paper.
- The pseudocode in Algorithm 1 seems to suggest the algorithm is exponential at first glance. The argument of having polynomial time complexity is weak given that $p$ is unknown in Theorem 1. For a simple transformer, that may not be an issue. However, issues could arise in modern-day transformer architecture with complex tasks as suggested by the performance in the model named Pythia-14m.
- Figure 2 seems to suggest the performance of the proposed algorithm may also depend on the task for the model, based on the difference in performance between language models and vision models. This gap is not well addressed in the paper.

**Questions:**

- What is self-repair in line 67?
- Why is argmax emphasized in the conditions of 4a and 4b?
- In condition 4b, it seems to suggest the causal node set must include all the downstream nodes of V along P, but why should that be the case? Why can $V$ not be a causal node set even when condition 4a is satisfied and $V$ partially satisfies condition 4b for the set of nodes in $P$ that are ancestors of $V$? Is it because it will not be enough for causal referencing?
- In lines 129-130, it says ‘any causal node set must have at least one parent node set that is also causal’. What is the definition of a parent node set? Does the parent node set also include the causal node set in line 129 in order to be causal? If not, it seems contradictive to the condition 4b for the following reason: let $V$ be the causal node set and $Pa(V)\setminus V$ be the parent node set, by definition of causal node set, $V$ must satisfy the condition 4a, but it contradicts with $Pa(V)\setminus V$ being causal by condition 4b.
- How are definitions 4a and 4b related to definition 6? Do they need to be stated separately?  Does Definition 6 imply satisfaction of conditions 4a and 4b in Definition 4?
- Regarding definition 6, why is there a need for changing graphical structures in order to be qualified as a sufficient intervention?
- Lines 170-171 claim that it requires identifying the causal node sets from the decision in order to identify which structures within the transformer contribute to the decision as causal paths. Is there proof of this claim?
- Why is it a good idea to treat input-dependent statistics, mean, and variance as fixed in line 191?
- Based on lines 5-7 in Algorithm 1, doesn’t it mean that minimality-based causal subset search per block is still an exponential search?
- How come Pythia-1b has a polynomial bound of $p$ higher than that of GPT2-xs even when Pythia-1b is a larger model?

---

> ### Author Response · Authors · 2025-11-21
>
> We sincerely thank the reviewer for the valuable and constructive feedback, which helped improve the clarity and rigor of our paper. Below, we provide our responses to each comment, along with the corresponding revisions made in the manuscript. Although some definition numbers have been updated in the revised manuscript, we retain the original numbering only in the response titles (e.g., [R1]~) to ensure easy mapping to the reviewer’s comments. Furthermore, we note that the main manuscript and the appendix (which were originally submitted as separate files) have now been merged into a single file.
>
> ---
>
> # [R1] About Targeting the Argmax Output
>
> This is because prior related work has primarily focused on classification tasks; however, we agree with the reviewer’s point and clarified in the main manuscript (line 66) that **our target is the model’s predicted class (i.e., the decision) in a classification setting.** Therefore, this also explains why we focus on the argmax when identifying causal paths for a decision: in classification tasks, the model’s prediction is determined by the argmax over the output logits.
>
> ---
>
> # [R2] About Def 4 (Causal Node set) and Property 1 (Causal Edge)
>
> ## [R2-1] On the ordering of definitions
>
> We agree with the reviewer that defining causal node set after causal path made the presentation difficult to follow. We adjusted the ordering in the main manuscript so that causal node sets are introduced before causal paths.
>
> ## [R2-2] On Lines 128–131
>
> First, we agree with the reviewer’s comment, and for clarification, we added explicit descriptions of edge directions and the notions of child and parent nodes within the definition of “Transformer as a Causal Graph.”
>
> Concretely, if a child node has three parent nodes, $v_1, v_2, v_3$, then the number of possible parent node sets is seven (all non-empty subsets of {$v_1, v_2, v_3$}). This reflects the fact that we **must account for potential interactions for the child node among parent nodes** when assessing causality; hence, parent node sets must be treated as distinct causal candidates.
>
> Furthermore, **under the directed acyclic graph condition** in the definition of “Transformer as a Causal Graph,” we emphasize that $Pa(V)$ and $V$ are strictly disjoint. Therefore, **the contradiction suggested by the reviewer does not arise.**
>
> ## [R2-3] On Lines 135–136
>
> We agree with the reviewer and **removed the phrase “possibly not minimal”**, which was indeed a source of confusion. Our intention in originally including this wording was to illustrate that if a child node has three parent nodes $v_1, v_2, v_3$, then the full parent node set {$v_1, v_2, v_3$} always satisfies both the necessity and sufficiency conditions in the definition of “Causal Node Set”; but, because such the node set may fail to satisfy causal minimality, we used the term “possibly not minimal.” We acknowledge that a parent node set that is strictly not minimal should not be considered causal under our framework. Therefore, to avoid misinterpretation, we removed this phrase from the manuscript.
>
> ---
>
> # [R3] About Self-repair
>
> We would like to clarify that self-repair was not introduced for the first time at line 67. **Rather, we first mentioned it earlier at line 39 with a citation,** noting that a prior study had already shown that self-repair can cause failures in measuring the true causal effects for a decision. Our contribution is to demonstrate that **such self-repair predominantly occurs outside the causal path,** which is why we state that “the path contains information essential to the decision.”
>
> In addition, self-repair has been discussed since the first version of our manuscript: it is covered in **Appendix Sections G.2 through G.4,** and also **noted in Section 3.2 of the main text.** For improved clarity, we also added a **note on self-repair at line 39 as well.**

---

> ### Author Response · Authors · 2025-11-21
>
> # [R4] About Def 6 (Sufficient Intervention)
>
> ## [R4-1] On mentioning the term before providing the formal definition
>
> We acknowledge that referring to the term prior to its formal definition may have made the presentation harder to follow. To address this, we added a note at the point of first mention so that readers can directly locate the corresponding definition.
>
> ## [R4-2] Relation to the definition of “Causal Node Set"
>
> We clarify that the necessity and sufficiency conditions in the definition of “Causal Node Set” can be evaluated only after applying a sufficient intervention. Here, a sufficient intervention refers to an intervention that is well-formed enough to **qualify as a valid intervention in our causal graph**; it does **not imply that the sufficient intervention itself guarantees or makes the necessity or sufficiency conditions hold.** These conditions must still be assessed independently once the intervention is applied.
>
> ## [R4-3] Need for modifying the graphical structure
>
> In Pearl’s causal theory, **an intervention corresponds to modifying the directed acyclic graph by removing the targeted incoming edges of a node.** Because our framework is grounded in this theory, any operation that qualifies as an intervention must also modify the graph structure accordingly.
>
> ---
>
> # [R5] About Def 8 (Reliability of Causal Evaluation) and Justifications
>
> ## [R5-1] Regarding Definition 8
>
> Thank you for the suggestion. We incorporated that into the manuscript.
>
> ## [R5-2] Regarding the design choices in our definitions
>
> We incorporated the reviewer’s advice and **added two remarks (line 134-142)** under the definition of the Causal Node Set to clarify why this particular causal formulation is necessary in our framework.
>
> ---
>
> # [R6] About Algorithm
>
> ## [R6-1] On the concern that the time complexity depends on an unknown $p$
>
> We acknowledge the reviewer’s point that the complexity is not guaranteed to be polynomial in the worst case when $p$ is unknown. At the same time, we emphasize
> that **it becomes non-polynomial only when the boundary of $p$ is extremely small, and this boundary becomes even smaller for modern transformers.** Specifically, the non-polynomial time complexity occurs only when $p \le \frac{1}{2^n - 2}$, where $n=2H+2$ is the number of unfolded paths per block, determined by the number of heads $H$. **As modern transformers use larger $H$, the polynomial bound decrease, making the worst-case scenario unlikely.** Empirically, Figure 2 shows the measured $p=0.082$ for Pythia-14M, which is well above the theoretical polynomial bound of 0.016.
>
> ## [R6-2] Regarding Figure 2
>
> Figure 2 is not intended to compare the performance of the models, but to empirically validate that **the theoretically predicted time complexity remains polynomial in practice.** Different transformers with distinct training recipes store information across different internal structures, leading to natural variation in how many internal nodes are causally relevant for a given decision. Thus, **differences across models reflect architectural or training differences, not task-specific trends.**
>
> The purpose of this figure is to provide empirical support that, **even though $p$ is unknown, values of $p$ that fall into the non-polynomial region are rarely observed in practice.** This justifies our claim that the algorithm runs in polynomial time on average.
>
> ## [R6-3] On whether Lines 5–7 of Algorithm 1 still make the algorithm exponential
>
> We agree that the outer loop alone would lead to exponential complexity. However, Line 6—**where causal minimality is checked and supersets of already-identified causal node sets are pruned**—is precisely what prevents exponential blow-up. As the subset size increases (Line 4), previously discovered causal node sets allow Line 6 to prune large portions of the search space. As shown in Theorem 1 and its proof (Appendix F.1), when **$p$ is small, the pruning induced by causal minimality ensures that the expected time complexity converges to polynomial.** This mechanism is the key reason the algorithm avoids exponential behavior in practice.

---

> ### Author Response · Authors · 2025-11-21
>
> # [R7] About Algorithm
>
> On the inconsistency between Equation (2) and the derivation in the Appendix
>
> We thank the reviewer for the careful check. We confirmed that a term mentioned in Equation (2) had been omitted in the derivation, and we **corrected this inconsistency** in the revised manuscript.
>
> ## [R7-1] Regarding Lines 170–171
>
> We believe that **a formal proof is not required at this point, since the statement follows directly from our definitions.** In our framework, causal paths are defined from causal node sets, causal node sets are defined in terms of their contribution to the decision, and each node in a node set corresponds to a path node that represents a specific internal structure in the transformer (i.e., the residual, attention, or MLP). Under these definitions, **the claim is a natural consequence of the framework.** Nevertheless, to make this connection clearer for readers, we **added an overview paragraph at the beginning of Section 2** that summarizes our causal framework and the overall tracing procedure.
>
> ## [R7-1] On the “fixed input-dependent statistics” in Line 191
>
> These terms, like softmax and GeLU, involve non-linear operations. Analogously to how we handle softmax and GeLU with the input-dependent statistics $D_{\alpha}$ and $D_{\beta}$, we treat these quantities as precomputed input-dependent statistics that can be factored into linear forms. **This allows us to interpret the transformer block as a sum of additive path terms.** During implementation, for each input sample, we first perform the full forward pass within a block and cache these statistics in memory, and then reuse them when constructing the path-wise decomposition.
>
> ## [R7-2] On why Pythia-1b has a larger polynomial bound on $p$ than GPT2-xs
>
> As clarified in Appendix Section G.1, the **Pythia family adopts a parallel residual structure** in which the block input is directly fed into both the attention and MLP components. As a result, each block in **Pythia has $n=H+2$ paths,** rather than $n=2H+2$ as in the standard residual design. For **Pythia-1b,** the number of heads is $H=8$, so the **corresponding polynomial bound of $p$ is $\frac{1}{2^{10} - 2}$**. In contrast, **GPT2-xs** follows the standard structure with $n=2H+2$ and $H=6$, leading to a **polynomial bound of $p$ as $\frac{1}{2^{14} - 2}$**  Therefore, due to this structural difference, Pythia-1b indeed has a larger (less restrictive) polynomial bound of $p$, as reflected in our analysis.

---

### Official Review · Reviewer_Kn3T · 2025-11-01

**Soundness:** 2
**Presentation:** 2
**Contribution:** 3
**Rating:** 4
**Confidence:** 3

**Summary:**

The paper introduces a method for identifying the causal paths that lead to an output decision. Specifically, it decomposes the transformer architecture into multiple traces (paths) and applies interventions to determine which paths actually cause the decision. The authors propose a concrete algorithm and analyze its efficiency both theoretically and empirically.

**Strengths:**

1. The paper provides analytical results on decomposing the transformer layer into multiple traces, which appears to be novel to my knowledge.
2. The paper tries to employ the actual causality framework (Halpern & Hitchcock 2011, Halpern 2015) to provide an interpretation of the machine learning framework (transformer).
3. The paper provides both a theoretical analysis of the time complexity for finding causal paths and experimental results demonstrating the algorithm’s efficiency in practice.

**Weaknesses:**

Some key definitions are either missing or introduced in an unclear order, which makes the paper difficult to follow and verify. Specifically,
1. pg2 Def 1. It seems that the authors want to provide a general definition here. However, this definition is unclear. In particular, the definition of "internal component" is missing. One would naturally think of internal components as tensor operators in a neural network, and this is not the authors' purpose. I couldn’t see how the causal graphs are structured until Section 2.3, so I suggest removing this definition or moving it to a later section.
2. pg2 Def 4. "subpath reference" has not been defined yet, which caused a lot of confusion. I also have some questions regarding this definition, which I included in the Questions section.
3. pg3 Def 5. This definition is quite confusing. "consecutively connected downstream node sets" is undefined. Also, please provide examples for "subpath reference" and "causal subpath reference".
4. pg3 Property 1. "child node" and "parent node sets" are undefined. Again, offering an example would be helpful. Also, does the result hold under any parameterizations (weights) in the transformer? Why not call it a proposition/theorem and provide a proof?
5. pg3 Def 6. The interplay between condition (6b) and Property 1 is not clear to me. How could an intervention produce a causal graph that violates Property 1? For condition (6a), does it basically require V to have a different set of parents? Condition (6c) is also quite confusing -- I can't quite understand its role here.
6. pg4 Equation (1). Missing definitions for $L_{ia}, L_{oa}$, etc.
7. pg5 Remark 2. The proof for average-case time complexity being polytime is missing.
8. pg6 Theorem 2. Is "Reliability" a new notion introduced by the authors or an existing notion? If it exists, please cite properly. Otherwise, I don't think it is acceptable to put the definition in the Appendix without justifying its validity and importance.

**Questions:**

1. pg2 Def 4. I checked out the two papers mentioned above, and I didn't see how Def 4 here is identical to the original definition. For 4(a), the original definition in (Halpern & Hitchcock 2011) seems to allow $\hat{V}'$ to have the same value as $\hat{V}$. For 4(b), the original definition also intervenes on the initial values of $V$ in addition to $\hat{V}'.$ Please let me know if I missed anything.
2. pg4 Eq (2). Why include these bias terms ($b_{attn}$, etc.) as paths if they do not depend on $Z_{ib]$?
3. pg5 Algorithm 1. Can you actually specify $\hat{V}$ as input to the algorithm without knowing the specific subset $V$?
4. Is the trick to rewrite the non-linear functions as Hadamard products something known? If so, please cite. Otherwise, please demonstrate.

---

> ### Author Response · Authors · 2025-11-21
>
> We sincerely thank the reviewer for the valuable and constructive feedback, which helped improve the clarity and rigor of our paper. Below, we provide our responses to each comment, along with the corresponding revisions made in the manuscript. Although some definition numbers have been updated in the revised manuscript, we retain the original numbering only in the response titles (e.g., [R1]~) to ensure easy mapping to the reviewer’s comments. Furthermore, we note that the main manuscript and the appendix (which were originally submitted as separate files) have now been merged into a single file.
>
> ---
>
> # [R1] About Def 1 (Transformer as Causal Graph)
>
> We agree that our original phrasing was unclear, which made it difficult for readers to understand how the transformer is structured as a causal graph. However, we hope the reviewer understands that interpreting the transformer as a causal graph must come first, since this foundational perspective underlies all subsequent definitions and causal analyses; **instead, to improve clarity for readers, we explicitly specified which internal components our method uses (line 96)**, rather than describing what could in principle serve as internal components, and **we added an overview paragraph at the beginning of Section 2** to help readers follow our causal framework and the tracing procedure.
>
> ---
> # [R2] About Def 4 (Causal Node Set)
>
> We acknowledge that introducing the term “subpath reference” before defining it may have hindered clarity. We therefore **revised the order of definitions accordingly.**
>
> We would like to answer the two questions regarding the relationship to the definition in Halpern & Hitchcock (2011).
>
> ## [R2-1] Relationship to Halpern & Hitchcock (2011)
>
> The conditions in our definition of a causal node set correspond directly to the conditions for actual cause in Halpern & Hitchcock (2011, p. 8). Specifically:
>
> - Our necessity condition corresponds to AC2(a).
> - Our sufficiency condition corresponds to AC2(b).
> - Our causal minimality condition corresponds to AC3.
>
> The reason AC1 does not appear explicitly in our definition is that, within a transformer, each internal node always possesses its value. Thus, AC1 is trivially satisfied and was omitted to avoid redundancy.
>
> ## [R2-2] Regarding whether $\hat{V}'$ and $\hat{V}$ may take the same value
>
> We would like to clarify that **both our definition and the original definition do not allow an intervention to assign the same value.** An intervention must fix the variable to a different value in order to reveal a counterfactual effect; assigning the original value cannot demonstrate any causal influence. This is consistent with Halpern & Hitchcock (2011), which states (p. 9, mid):
>
> > “AC2(a) is essentially the standard counterfactual … if $\vec{X}$ had a different value, then $\phi$ would have been false.”
> >
>
> This statement makes it clear that interventions require assigning different values.
>
> ## [R2-3] Regarding intervention on the initial value of $V$"
>
> We understand that the reviewer is referring to $\vec{Z'}$ in the notation used by Halpern & Hitchcock (2011). We would like to clarify that, **in both our and original definition, the intervention is not applied to $P$** (equal to $\vec{Z'}$ in Halpern & Hitchcock (2011)), nor even to $V$. **These nodes are perturbed, not intervened on.** Perturbation means that the node is not forcibly set to a new value (i.e., not remove outgoing edges); rather, its value may change indirectly due to interventions applied to its parent nodes.
>
> For example, let $E1$ be "tree catches fire," $E2$ be "throwing a lit match onto the tree," $E3$ be "strong wind," and $Y$ be "tree burns down." Here, suppose that $E1$ is a cause and that the paths to $Y$ are modeled as $E2\rightarrow E1\rightarrow Y$ and $E3\rightarrow E1\rightarrow Y$.
> If we intervene on $E3$ (our $\hat{V'}$), i.e., remove the wind, to test whether $E2$ (our $V$) is a cause, then $E1$ (our $P$) may occur at a different time or intensity **(i.e., it is perturbed)**, but $Y$ still happen. This shows the discussion in Halpern & Hitchcock (2011, p. 9, bottom):
>
> > "Moreover, when $\vec{X}=\vec{x}$, although the values ... still be true, as long as $\vec{X}=\vec{x}$"

---

> ### Author Response · Authors · 2025-11-21
>
> # [R3] About Def 5 (Causal Referencing)
>
> To avoid confusion, we provided a downstream node set (line 107) in the definition of “Transformer as Causal Graph”, and we removed the term "consecutively connected" when defining a subpath reference. At the same time, we refined the title of the definition from "Causal Referencing" to "Causal Subpath Reference and Causal Referencing" for clarity.
>
> As an illustrative example, we refer to Example 1 in the appendix. Suppose that, in the final block, the node set $V_1^{(2)}$ is already identified as causal and $V_2^{(2)}$ is not. Then, when evaluating whether $V_1^{(1)}$ in the first block is causal, both $V_1^{(2)}$ and $V_2^{(2)}$ from the final block serve as subpath references for $V_1^{(1)}$; among them, only $V_1^{(2)}$ qualifies as a causal subpath reference.
>
> ---
>
> # [R4] About Property 1 (Causal Edge)
>
> We added explicit descriptions (line 105-106) of edge directions and the notions of child and parent nodes within the definition of “Transformer as a Causal Graph.” Referring again to Example 1 in the appendix (as discussed in [R3]), note that $V_1^{(2)}$ and $V_2^{(2)}$ are the children of $V_1^{(1)}$, and conversely, $V_1^{(1)}$ is the parent of both node sets.
>
> Importantly, we clarify that **this property does not rely on any particular parameterization of the model.** Instead, it arises purely from the structural characteristics of transformers that satisfy the conditions of Definition 2. We also updated the property of "Causal Edge" to reflect this clarification.
>
> ---
>
> # [R5] About Def 6 (Sufficient Intervention)
>
> **Property 1 naturally holds under our causal framework** because it follows directly from the structural characteristics of transformers. Except for token inputs, every internal nodes in a transformer always have at least one parent; hence no node is parentless or unobserved, satisfying causal sufficiency in the definition of "Transformer as Causal Graph." Furthermore, under Property 1, every child node must have at least one causal parent. Consequently, if all incoming edges from every parent of a node were removed, the decision must change—mirroring the fact that removing all causes should necessarily change the effect.
>
> To illustrate this intuition, consider a simplified world where a tree burns only if there is oxygen and contact with fire (our causal framework). If we eliminate both conditions (i.e., remove all edges via intervention), the tree should never burn; **spontaneous combustion does not exist in this world.** This directly reflects Property 1.
>
> However, as discussed in Appendix Section D.2, certain interventions—such as Zero Masking, which sets all parent node sets of a block to zero—**break this causal structure.** When every parent is zeroed out, the classifier may output the same label purely due to its bias vector. In this case, regardless of the token inputs, the decision remains unchanged even after all causal edges are removed, **akin to “spontaneous combustion”** in our analogy. Such behavior lies outside the causal framework we define. Therefore, if one wishes to use an intervention that fails to satisfy our conditions of sufficient intervention (e.g., zero masking), **that intervention implicitly assumes a causal framework fundamentally different from ours.**
>
> **Additional clarification** regarding Causal Structural Isomorphism and Intervention Controllability for sufficient intervention is provided separately in our response to **[R12].**
>
> ---
>
> # [R6] About Equation 1
>
> We added these definitions to the main manuscript.
>
> ---
>
> # [R7] About Remark 2 (Expected Time Complexity of Minimality-based Subset Search)
>
> The proof is provided in the Appendix Section F.1. In addition, the reason why Remark 2 only discusses the cases where $p=1$ or $p$ is very small ($p \le \frac{1}{2^n - 2}$) is that **these are precisely the regimes in which non-polynomial behavior may arise; once $p$ exceeds this range, the expected complexity becomes polynomial.**

---

> ### Author Response · Authors · 2025-11-21
>
> # [R8] About Reliability in Theorem 2
>
> Prior work has used reliability without a formal definition, or has evaluated it without considering all possible paths; thus, the notion is newly introduced by our work. We agree with the reviewer’s comment and added the corresponding clarification to the main manuscript.
>
> ---
>
> # [R9] About Equation 2
>
> Although these components do not directly depend on $Z_{ib}$, they are still part of the mechanisms that contribute to the final decision, and therefore must be included in the set of paths. Excluding them would make it **impossible to reconstruct the original decision**, meaning that one cannot claim to consider all paths. Moreover, **these components operate jointly with either the Attention or MLP** computations inside the transformer block; thus, we assign them to the corresponding paths in which they function.
>
> ---
>
> # [R10] About Algorithm 1
>
> **$V$ is not unknown;** it denotes a subset of $V_p$ as described in Line 5 of Algorithm 1. For example, when $s=1$, the subsets are {$v_1$}, {$v_2$}, …, and each of these becomes a concrete instance of $V$ during iteration. When $s=2$, the subsets are {$v_1, v_2$}, {$v_1, v_3$}, …, and the algorithm iterates over all such combinations (while the condition in Line 6 controls the overall complexity).
>
> Thus, in each iteration, $V$ is fully instantiated as a specific subset of $V_p$, and all components outside $V$ and $P$ are set to $\hat{V}$.
>
> ---
>
> # [R11] About Trick to Rewrite Non-Linear Function
>
> This computational trick is also, to the best of our knowledge, **introduced for the first time in our work.** To decompose the transformer block into additive path terms, the main bottleneck lies in rewriting softmax and GeLU into forms that can be expressed as additions or multiplications. However, since non-linear functions cannot in general be converted into linear forms, we instead rewrite them as element-wise products. The corresponding scaling factors can only be obtained from the outputs of the non-linear functions themselves. During implementation, for each input sample, we first perform the full forward pass within a block, **store the outputs of softmax and GeLU in memory, and then divide them element-wise by $z$,** computed as $D_{\alpha}=\textup{softmax}(z/\sqrt{d_h})/z, D_{\beta}=\phi(z)/z$. We added this explanation to the main manuscript.

---

> ### Author Response · Authors · 2025-11-21
>
> # [R12] Example for sufficient intervention
>
> ## [R12-1] For Causal Structural Isomorphism
>
> **(Setup)**
>
> Consider the toy model consisting of a single transformer block with a single attention head:
> Let $z_{token}\in \mathbb{R}^{T\times d_m}$ denote the original token embeddings of length $T$. As described in Equation (7) of the main paper, consider the unfolded representation at the output of the first block (denoted $z_{ob}$), where $z_{ib}=z_{token}$ is the block input. Then, the output of the block can be expressed as:
> $z_{ob} = z_{token}+\textup{[Attention Only]}+\textup{[Attention+MLP]} + z_{token} W_{ln_2}^\top W_{im}^\top D_\beta W_{om}^\top + b_{mlp}$ .
>
> Here, the attention-related terms (i.e., [Attention only] and [Attention+MLP]) contain $z_{token}$ through $z_q^{(h)}, z_k^{(h)}, z_v^{(h)}$, but for brevity of expression, we refer to them by name only. Consequently, $z_{ob}$ passes through the classifier, parameterized by $W_{cls}$ and $b_{cls}$, resulting in the final output $y=z_{ob} W_{cls}^\top +b_{cls}$.
>
> Each addictive term in the above is equal to a distinct path node, and aligned with the path node set $V_p=[v_1, v_2, v_3, v_4]$ in the input of Algorithm 1 of the main paper as:
> $v_1 = z_{token}$
>
> $v_2 = \textup{[Attention Only]}$
>
> $v_3 = \textup{[Attention+MLP]}$
>
> $v_4=z_{token} W_{ln_2}^\top W_{im}^\top D_\beta W_{om}^\top + b_{\text{mlp}}$
>
> **(Scenario)**
>
> Consider two subsets of path nodes as evaluation targets: $V_1=[v_1,v_2,v_3]$ and $V_2=[v_2,v_3,v_4]$. For each subset, non-target nodes (i.e., nodes not in the subset) is intervened on to isolate the causal effect of the subset. Specifically, for $V_1$, $v_4$ is replaced by $v_4'$; for $V_2$, $v_1$ is replaced by $v_1'$. This yields the corresponding intervened graphs for each case.
>
> **TOKEN RESAMPLING** substitutes the original token embedding $z_{token}$ with a different token embedding $z_{token}'\in \mathbb{R}^{T\times d_m}$, drawn from the model’s vocabulary. This yield:
>
> - $v_4'=z_{token}' W_{ln_2}^\top W_{im}^\top D_\beta W_{om}^\top + b_{mlp}$
> - $v_1'=z_{token}'$
>
> **DIRECT NOISE** adds Gaussian noise (i.e., $z_{noise}\in \mathbb{R}^{T\times d_m}$) to “the target node”. This yield:
>
> - $v_4'=v_4+z_{noise}$
> - $v_1'=v_1+z_{noise}$
>
> This illustrates that **TOKEN RESAMPLING** preserves causal structural isomorphism between the computation graph and its mathematical expression, whereas **DIRECT NOISE** violates it.
>
> Under TOKEN RESAMPLING:
>
> - The intervened graph for $V_1$ has the expression: $z_{token}+v_2+v_3+z_{token}' W_{ln_2}^\top W_{im}^\top D_\beta W_{om}^\top + b_{\text{mlp}}$
> - while the graph for $V_2$ has: $z_{token}'+v_2+v_3+z_{token} W_{ln_2}^\top W_{im}^\top D_\beta W_{om}^\top + b_{\text{mlp}}$
>
> These two expressions are clearly distinguishable, since they are grounded in two distinct root variables, $z_{token}$ and $z_{token}'$. Moreover, even beyond this specific example, such distinct graphs can always be distinguished in this way. Thus, there exists a one-to-one correspondence between the intervened graph and its mathematical representation, satisfying Definition 5.a.
> In contrast, under DIRECT NOISE:
>
> - The intervened graph for $V_1$ becomes: $v_1+v_2+v_3+v_4+z_{noise}$
> - and for $V_2$: $v_1+z_{noise}+v_2+v_3+v_4$
>
> Due to the commutativity of vector addition, these two expressions are algebraically indistinguishable. As a result, two different intervention targets yield the same mathematical form, violating the one-to-one mapping required by structural isomorphism.
>
> ## [R12-2]  For Intervention Controllability
>
> **(Setup & Scenario)**
>
> Consider the above setup and scenario with the subset $V_1$.
> **TOKEN RESAMPLING** substitutes $v_4$ in the same way as described above.
>
> **NOISE TOKEN** adds Gaussian noise (i.e., $z_{noise}\in \mathbb{R}^{T\times d_m}$) to “the original token embedding $z_{token}$," and uses the resulting noisy embedding to replace the target node. This yield:
>
> - $v_4'=(z_{token}+z_{noise}) W_{ln_2}^\top W_{im}^\top D_\beta W_{om}^\top + b_{mlp}$
>
> This demonstrates that **TOKEN RESAMPLING** operates within the model’s in-distribution, whereas **NOISE TOKEN** may yield out-of-distribution representations due to the added noise term.
>
> In more detail, under TOKEN RESAMPLING, the substituted embedding $z_{token}'$ is drawn from the model's own vocabulary embedding space. The resulting representation $v_4'$, computed downstream from $z_{token}'$, therefore remains on the model's learned manifold. So, this method satisfies Definition 5.c.
>
> In contrast, under DIRECT NOISE, the introduced noise term $z_{token}$ leads to an addictive component in the form of $z_{noise} W_{ln_2}^\top W_{im}^\top D_\beta W_{om}^\top$, which cannot be guaranteed to produce an in-distribution representation. Moreover, in common models beyond this toy example, the scale of this noise term can be excessively amplified or diminished across blocks, leading to out-of-distribution representations that violate Definition 5.c.

---

### Author Response · Authors · 2025-11-30
**Author Revision Summary (for clarity of the revised manuscript)**

Before summarizing the revisions, we first highlight the major strengths consistently recognized across reviewers:
- Addresses **a core issue in modern AI, causal interpretability** (Kn3T, a9Ri, BTiY), in a **clear and well-written** manner (a9Ri, BTiY).
- Provides **an insightful path-level decomposition of transformer layers** (Kn3T, Ruuc) and an **efficient method for causal path tracing**, supported by theoretical guarantees (Kn3T, BTiY).
- Demonstrates that the proposed algorithm is **efficient in practice through empirical evaluations** (Kn3T, Ruuc, BTiY).
- Offers **empirical findings** such as **self-repair occurring outside causal paths** and **class-wise shared causal paths in image classification**, which reviewers highlighted as valuable (Kn3T, Ruuc, a9Ri, BTiY).

---
Across the reviews, most concerns centered on the ordering of definitions and clarifying certain ones, **rather than on the methodology, theoretical guarantees, or empirical findings.** To address these issues, we have substantially improved the presentation in the revised manuscript. **Key revisions** include:

- **Reorganized Section 2**, including a clearer ordering of definitions and a new overview paragraph.
- **Clarified edge directions in Definition 2** and added explicit definitions of parent/child.
- Added a **do-notation–based formulation to Definition 4.**
- Added **Remarks 2 and 3** to justify design choices regarding causal node sets and intervention constraints.
- **Moved Definition 7 (reliability)** from the appendix to the main text for easier reference.
- **Removed inaccurate wording,** specifically the phrases "possibly not minimal" in Property 1 and "NP-complete" in the main text.
- **Fixed the missing term in Equation (2).**

We hope these revisions improve the clarity of the causal framework and make the overall contributions of the paper easier to follow.

---

### Meta-Review · Area_Chair_W9EV · 2026-01-11

**Summary:**

The paper proposes Causal Path Tracing, a framework to trace decision-relevant internal computations in transformers. The approach (i) unfolds each transformer block into a causal graph whose nodes correspond to additive “path” terms, (ii) defines causal node sets using necessity/sufficiency/minimality-style criteria, and (iii) uses a minimality-based subset search to identify within-block causal node sets/paths, claiming polynomial-time complexity on average under an expectation model. It further proposes a union-based reference strategy for tracing across blocks. Experiments aim to demonstrate practical runtime, reduced “self-repair” along identified causal paths, and that traced paths can recover the model decision and show class-wise path patterns.

### Strengths
- Targets an important problem: scalable causal/path-based interpretability for transformers (Kn3T, a9Ri, BTiY).
- Provides a concrete decomposition of transformer blocks into path terms and an explicit search procedure intended to enumerate decision-relevant paths (Kn3T, Ruuc).
- Empirical observations (e.g., self-repair less frequent along traced paths; class-wise/shared path patterns) are potentially interesting for interpretability analyses (Ruuc, a9Ri, BTiY).

### Weaknesses
- Clarity and rigor issues are substantial and recurring across reviewers: key notions (causal node sets, path/node-set connectivity, parent/child relations, sufficient intervention, reliability) were originally introduced out of order or loosely, making the framework difficult to verify (Kn3T, Ruuc, BTiY). While the rebuttal adds clarifications, at least one reviewer reports the “vague definitions” concern remains.
- The algorithmic/theoretical positioning is not fully convincing as presented: the average-case polynomial-time claim depends on an unknown parameter p and an assumption model that reviewers found too weak/general to support strong conclusions for modern transformers (Ruuc, BTiY). The pseudocode also appears exponential without careful reading of pruning assumptions (Ruuc).
- The submission contained technical/expository inaccuracies that required correction (e.g., “NP-complete” phrasing; confusing “possibly not minimal”; an inconsistency/missing term in Eq. (2)), which undermines confidence in the overall mathematical presentation (Ruuc, BTiY).

The negative review raises concerns that go beyond minor nitpicks: (i) the core complexity story relies on an unknown p and a weak expectation model, and (ii) multiple foundational definitions/claims were unclear enough to impede verification. Given that the claimed guarantees and causal formalization are central to the paper’s contribution, lingering ambiguity in these parts materially affects confidence in correctness and reproducibility of the proposed causal tracing framework. Should these be resolved in the next version, I would be more than happy to see it accepted.

**Reviewer Concerns:**

Addressed:
- Reordered/clarified definitions; added explicit edge directions and parent/child notions; added examples (Kn3T, Ruuc).
- Fixed the missing term / inconsistency in Eq. (2) (Ruuc).
- Removed inaccurate wording (“NP-complete”; “possibly not minimal”) and clarified intent (Ruuc, BTiY).
- Added a do-notation style restatement for part of the causal node set definition and clarified reliability by moving its definition into the main text (Kn3T, BTiY, Ruuc).
- Clarified Algorithm 1 iteration and explained pruning/minimality’s role; pointed to an appendix proof for expected complexity (Kn3T, Ruuc).

Still outstanding:
- Residual lack of clarity/rigor in the causal formalization and path/node-set definitions: reviewer BTiY explicitly maintained their score after rebuttal, stating that vagueness issues remain and are shared by others.
- The practical meaning/robustness of the average-case polynomial-time claim remains uncertain for the setting emphasized by the paper, since p is unknown and the expectation model is a key assumption; reviewer Ruuc’s central skepticism here is not fully resolved by clarifications alone.
- Limited demonstrated downstream utility beyond interpretability phenomena; applications are stated as future directions rather than evidenced (a9Ri).

**Reviewer Scores:**

- Reviewer Kn3T (rating 4, confidence 3): Likely small increase (4 → 5) due to addressed definition-ordering and missing-definition issues, though remaining concerns about verification may persist.
- Reviewer Ruuc (rating 2, confidence 3): Likely small increase (2 → 3) given that some concrete errors were fixed (Eq. (2), confusing phrasing), but core concerns about weak complexity argument/p-dependence and overall writing/justification likely keep the score in reject territory.
- Reviewer a9Ri (rating 6, confidence 2): Likely unchanged (6 → 6). Reviewer indicated concerns were addressed but deferred to others.
- Reviewer BTiY (rating 4, confidence 3): Unchanged (4 → 4), explicitly stated after rebuttal.

---

### Decision · Program_Chairs · 2026-01-26

Reject